# Private Regression via Data-Dependent Sufficient Statistic Perturbation

**Cecilia Ferrando**  *cferrando@cs.umass.edu*
*Manning College of Information and Computer Sciences*
*University of Massachusetts Amherst*

**Daniel Sheldon**  *sheldon@cs.umass.edu*
*Manning College of Information and Computer Sciences*
*University of Massachusetts Amherst*

**Reviewed on OpenReview:** *https://openreview.net/forum?id=gtCfDKm9ME*

## Abstract

Sufficient statistic perturbation (SSP) is a widely used method for differentially private linear regression. SSP adopts a data-independent approach where privacy noise from a simple distribution is added to sufficient statistics. However, sufficient statistics can often be expressed as linear queries and better approximated by data-dependent mechanisms. In this paper we introduce data-dependent SSP for linear regression based on post-processing privately released marginals, and find that it outperforms state-of-the-art data-independent SSP. We extend this result to logistic regression by developing an approximate objective that can be expressed in terms of sufficient statistics, resulting in a novel and highly competitive SSP approach for logistic regression. We also make a connection to synthetic data for machine learning: for models with sufficient statistics, training on synthetic data corresponds to data-dependent SSP, with the overall utility determined by how well the mechanism answers these linear queries.

## 1 Introduction

Differential privacy (DP) (Dwork et al., 2006) is an established mathematical framework for protecting user privacy while analyzing sensitive data. A differentially private algorithm injects calibrated random noise into the data analytic process to mask the membership of single records in the data, limiting the information revealed about them in the output of the privatized algorithm. The literature encompasses numerous methods for achieving differential privacy across a wide range of machine learning algorithms, including objective perturbation (Chaudhuri and Monteleoni, 2008; Chaudhuri et al., 2011; Kifer et al., 2012; Jain and Thakurta, 2013), with applications to models trained via empirical risk minimization; gradient perturbation (Bassily et al., 2014; Abadi et al., 2016), which is commonly used in deep learning and models trained via gradient descent; one-posterior sampling (Wang et al., 2015; Dimitrakakis et al., 2017) with applications in private Bayesian inference; and finally, sufficient statistic perturbation (SSP) (Vu and Slavkovic, 2009; McSherry and Mironov, 2009; Dwork and Smith, 2010; Zhang et al., 2016; Foulds et al., 2016; Wang, 2018; Bernstein and Sheldon, 2019; Ferrando et al., 2022), with natural applications in exponential family estimation and linear regression.

SSP adds calibrated random noise to the sufficient statistics of the problem of interest and uses the noisy sufficient statistics downstream to retrieve the target estimate. It is appealing for a number of reasons. Sufficient statistics are by definition an information bottleneck, in that they summarize all the information about the model parameters (Fisher, 1922). For many models, like linear regression and exponential family distributions, their sensitivity is easy to quantify or bound, simplifying the DP analysis. Finally, they can be privatized via simple additive mechanisms, like the Laplace or Gaussian mechanism (Dwork et al., 2014).

Existing `SSP` methods are *data-independent*, meaning they add noise to the sufficient statistics in a way that does not depend on the underlying data distribution. In a different branch of DP research, recent work has shown that *data-dependent* mechanisms are the most effective for query answering and synthetic data (Hardt et al., 2012; Gaboardi et al., 2014; Zhang et al., 2017; Aydore et al., 2021; Liu et al., 2021; McKenna et al., 2022).

In this paper, we introduce `DD-SSP`, a data-dependent `SSP` method that leverages private linear query answering to release differentially private (DP) sufficient statistics. Its most immediate application is to linear regression, where finite sufficient statistics exist and, as we demonstrate, can be estimated privately through simple transformations of DP pairwise marginals. Furthermore, we extend the application of `DD-SSP` to models without defined finite sufficient statistics by proposing a novel framework for logistic regression, where approximate sufficient statistics are derived and released in a data-dependent way to train the model by optimizing an approximate loss function.

The proposed framework can be used with virtually any DP query answering algorithm. In this paper, we use `AIM` (McKenna et al., 2022) as our primary method, but show the overall method is robust to different choices. The main advantage of `AIM` is its demonstrated accuracy at preserving marginal queries, albeit at the cost of restriction to discrete data (a requirement in `AIM`). We show experimentally that `DD-SSP` outperforms the state-of-the-art data-independent `SSP` method `AdaSSP` (Wang, 2018) for linear regression, and for logistic regression tasks, `DD-SSP` achieves better results than the widely used objective perturbation baseline. We also compare `DD-SSP` with `DP-SGD` (Abadi et al., 2016), known to achieve excellent performance when hyperparameters are properly fine-tuned. Our results show that the proposed method is competitive with `DP-SGD` when the privacy cost of hyperparameter tuning is taken into account.

Finally, we elaborate on the significance of our results with respect to the increasingly popular practice of training machine learning models on DP synthetic data. Our results support the observation that for these models training on synthetic data generated by linear-query preserving mechanisms effectively corresponds to a form of data-dependent `SSP`.

## 2 Background

### 2.1 Differential privacy

Differential privacy (DP) (Dwork et al., 2006) has become the preferred standard for preserving user privacy in data analysis, and it has been widely adopted by private and governmental organizations. Differential privacy allows many data computations (including statistical summaries and aggregates, and the training of various predictive models) to be performed while provably meeting privacy constraints. The concept of neighboring datasets is integral to differential privacy, which aims to limit the influence of any one individual on the algorithmic output in order to safeguard personal privacy.

**Definition 2.1** (Neighboring datasets)**.** Two datasets $D$ and $D'$ are considered neighbors ($D \sim D'$) if $D'$ can be created by adding or deleting a single record from $D$.

Based on the concept of neighboring datasets, we can define the *sensitivity* of a function:

**Definition 2.2** ($L_2$ sensitivity)**.** Given a vector-valued function of the data $f : \mathcal{D} \to \mathbb{R}^p$, the $L_2$ sensitivity of $f$ is defined as $\Delta(f) = \max_{D \sim D'} \|f(D) - f(D')\|_2$.

Differential privacy can be achieved via different mechanisms of addition of calibrated random noise, with slightly different definitions. In this paper, we adopt $(\epsilon, \delta)$-DP, which can be achieved via the Gaussian Mechanism.

**Definition 2.3** (($\epsilon, \delta$)-Differential Privacy)**.** A randomized mechanism $\mathcal{M} : \mathcal{D} \to \mathcal{R}$ satisfies $(\epsilon, \delta)$-differential privacy if for any neighbor datasets $D \sim D' \in \mathcal{D}$, and any subset of possible outputs $S \subseteq \mathcal{R}$

$$\Pr[\mathcal{M}(D) \in S] \le \exp(\epsilon) \Pr[\mathcal{M}(D') \in S] + \delta$$

**Definition 2.4** (Gaussian mechanism)**.** Let $f : \mathcal{D} \to \mathbb{R}^p$ be a vector-valued function of the input data. The Gaussian mechanism is given by

$$\mathcal{M}(D) = f(D) + \nu$$

where $\nu$ is random noise drawn from $\mathcal{N}(0, \sigma^2 I_p)$ with variance $\sigma^2 = 2\ln(1.25/\delta) \cdot \Delta(f)^2/\epsilon^2$ and $\Delta(f)$ is the $L_2$-sensitivity of $f$. That is, the Gaussian mechanism adds i.i.d. Gaussian noise to each entry of $f(D)$ with scale $\sigma$ dependent on the privacy parameters.

**Proposition 2.5.** *(Dwork et al., 2014) The Gaussian mechanism in Definition 2.4 satisfies $(\epsilon, \delta)$-DP.*

**Proposition 2.6** (Post-processing property of DP)**.** *(Dwork et al., 2014) If $\mathcal{M}(D)$ is $(\epsilon, \delta)$-DP, then for any deterministic or randomized function $g$, $g(\mathcal{M}(D))$ satisfies $(\epsilon, \delta)$-DP.*

## 2.2 Data-independent sufficient statistic perturbation

Sufficient Statistic Perturbation (SSP) is a widely used differential privacy mechanism that introduces privacy noise at the level of summary statistics. For example, differentially private linear regression is commonly achieved by directly perturbing the sufficient statistics $X^T X$ and $X^T y$ via an additive noise mechanism, such as the Gaussian mechanism, where $X \in \mathbb{R}^{n \times d}$ is the feature matrix and $y \in \mathbb{R}^d$ is the response vector. The noise is added independently to each entry of these statistics based on the privacy parameters and the global sensitivity: the sensitivity of $X^T X$ is bounded by $\|\mathcal{X}\|^2$, and that of $X^T y$ by $\|\mathcal{X}\| \cdot \|\mathcal{Y}\|$, where $\|\mathcal{X}\|$ and $\|\mathcal{Y}\|$ are bounds on the feature and target vectors respectively (for a detailed derivation of the sensitivity bounds, see Appendix D). Specifically, SSP for linear regression is usually performed by splitting the privacy budget into $(\epsilon_1, \delta_1)$ and $(\epsilon_2, \delta_2)$ so that $\epsilon_1 + \epsilon_2 = \epsilon$ and $\delta_1 + \delta_2 = \delta$, and computing the following:

- $\widehat{X^T X} = X^T X + \nu Z_1$ where $Z_1 \in \mathbb{R}^{d \times d}$ is a symmetric matrix with upper-triangular entries sampled from $\mathcal{N}(0, 1)$, and the noise scale is $\nu^2 = \frac{2\ln(1.25/\delta_1)\|\mathcal{X}\|^4}{\epsilon_1^2}$.

- $\widehat{X^T y} = X^T y + \upsilon Z_2$ where $Z_2 \sim \mathcal{N}(0, I_d)$, and the noise scale is $\upsilon^2 = \frac{2\ln(1.25/\delta_2)\|\mathcal{X}\|^2\|\mathcal{Y}\|^2}{\epsilon_2^2}$.

The privatized estimates $\widehat{X^T X}$ and $\widehat{X^T y}$ are then used to compute the private estimator $\hat{\theta} = \left(\widehat{X^T X}\right)^{-1} \widehat{X^T y}$. This approach is simple and effective, but treats all entries in the sufficient statistics uniformly, adding entry-wise noise components calibrated to the same scale that only depends on the data through measures of global sensitivity. This implies that the noise mechanism does not exploit the structure of the underlying data to determine how the privacy noise is allocated. We call this mechanism data-*independent* SSP.

Later in this paper, we will instead leverage data-*dependent* privacy mechanisms for estimating sufficient statistics. Data-dependent mechanisms have gained traction in the field of differentially private data generation as the most effective mechanisms for query answering and synthetic data. The next section introduces the relevant background on differentially private synthetic data to introduce the context and motivation for our proposed data-dependent SSP methods.

## 2.3 Differentially private synthetic data and query-answering mechanisms

Differentially private synthetic data generation aims to produce surrogate data that preserves key statistical properties of the original data while ensuring privacy (Hardt et al., 2012; Zhang et al., 2017; Xie et al., 2018; Jordon et al., 2019; McKenna et al., 2019; Rosenblatt et al., 2020; Vietri et al., 2020; Liu et al., 2021; Aydore et al., 2021; McKenna et al., 2021b; Vietri et al., 2022; McKenna et al., 2022). Rather than perturbing the data or a downstream model directly, these methods fit a generative model to privatized statistics of the data, then sample from the model to produce surrogate data. This synthetic data can then be used for a wide range of analytical tasks without further privacy cost.

The following definitions allow us to formally elaborate on differentially private synthetic data.

**Definition 2.7** (Dataset)**.** A dataset $D$ is defined as a collection of $n$ potentially sensitive records. Each record $\chi^{(i)} \in D$ is a $d$-dimensional vector $(\chi_1^{(i)}, \dots, \chi_d^{(i)})$.

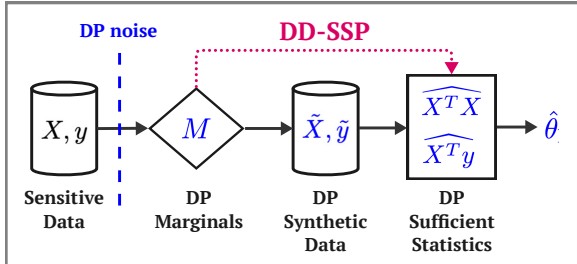

Figure 1: Diagram of the `SSP` data-independent workflow (left) vs the data-dependent linear query answering mechanism for marginal release/synthetic data workflow (right). Quantities indicated in blue follow privacy noise injection and are differentially private.

**Definition 2.8** (Domain). The domain of possible values for attribute $\chi_j^{(i)}$ is $\Omega_j = \{1, \ldots, m_j\}$. The full domain of possible values for $\chi^{(i)}$ is thus $\Omega = \Omega_1 \times \cdots \times \Omega_d$ which has size $\prod_j m_j = m$.

We will later talk about numerical encodings of attributes (Section 3.1).

**Definition 2.9** (Marginals). Let $r \subseteq [d]$ be a subset of features, $\Omega_r = \prod_{j \in r} \Omega_j$, $m_r = |\Omega_r|$, and $\chi_r = (\chi_j)_{j \in r}$. The marginal on $r$ is a vector $\mu_r \in \mathbb{R}^{m_r}$ indexed by domain elements $t \in \Omega_r$ such that each entry is $\mu_r[t] = \sum_{\chi \in D} \mathbb{1}[\chi_r = t]$ (i.e., counts).

With marginal queries, one record can only contribute a count of one to a single cell of the output vector. For this reason, the $L_2$ sensitivity of a marginal query $M_r$ is 1, regardless of the attributes in $r$. This facilitates the differential privacy analysis for marginal queries.

**Definition 2.10** (Workload). A marginal workload $W$ is defined as a set of marginal queries $r_1, \ldots, r_K$ where $r_k \subseteq [d]$.

The goal of workload-based synthetic data generation models to minimize the approximation error on workload queries.

A widely used framework in this space is the *select-measure-reconstruct* paradigm (Hay et al., 2009; Li et al., 2010; Ding et al., 2011; Xiao et al., 2012; Li and Miklau, 2012; Xu et al., 2013; Yaroslavtsev et al., 2013; Li et al., 2014; Qardaji et al., 2014; Zhang et al., 2014; Li et al., 2015; McKenna et al., 2021a), in which a workload of marginal queries is selected, privately measured, and then used to reconstruct either the full data distribution or a synthetic dataset. Many algorithms in this class are *data-dependent*: the choice of which queries to measure is based on the data itself, leading to better utility for a fixed privacy budget.

One example of select-measure-reconstruct method, and the one we use primarily in this paper, is `AIM` (McKenna et al., 2022). `AIM` (Adaptive and Iterative Mechanism) is a state-of-the-art data-dependent method that selects marginal queries to measure via a scoring function that assesses the expected utility of the measurements based on the data. This data-awareness allows `AIM` to focus the privacy budget where it matters most, depending on the data distribution. Note that `AIM` uses zero-concentrated differential privacy (zCDP) (Bun and Steinke, 2016), an alternative privacy definition; the Gaussian mechanism as defined in Section 2.1 satisfies $\frac{1}{2\sigma^2}$-zCDP (Bun and Steinke, 2016). In our experiments, we work with $(\epsilon, \delta)$-DP and the conversion to zCDP is handled internally in `AIM`.

In our experiments, we use `AIM` directly to release marginals without sampling synthetic data, treating it as a general-purpose, data-dependent linear query release mechanism.

## 3   Methods

In this paper, we propose methods to privately estimate sufficient statistics in a data-dependent way. Specifically, our methods leverage privately released marginals computed by a data-dependent linear query answering algorithm to estimate sufficient statistics. We call this set of methods `DD-SSP`. The main advantage of `DD-SSP` is that it is *data-dependent.* Our approach stems from the insight that, while data-independent noise addition is a simple and established approach to `SSP`, many sufficient statistics can be expressed as linear queries, creating an opportunity to improve utility by using data-dependent query-answering DP mechanisms, which often achieve higher accuracy than simple additive noise mechanisms (McKenna et al., 2022). In fact, using private synthetic data to train certain models *is already* a form of SSP. Figure 1 compares a standard `SSP` workflow and our proposed `DD-SSP` workflows. As seen in the figure, the pipeline of releasing synthetic data and then training a model via sufficient statistics can be viewed as a specific way of privatizing sufficient statistics for model training.

For linear regression, the application is straightforward: the problem has finite sufficient statistics and we demonstrate that two-way marginal queries are sufficient for their estimation. For other models, finite sufficient statistics are not available, but a polynomial approximation of the loss functions provides approximate sufficient statistics. This is the case for logistic regression, where finite sufficient statistics do not exist, but a Chebyshev polynomial approximation based on Huggins et al. (2017) allows us to propose an *approximate* version of the learning objective based on approximate sufficient statistics that can be expressed as linear queries, again retrievable via two-way marginal tables.

We use the synthetic data mechanism `AIM` as our private query answering algorithm and modify its implementation to output marginals directly, without the need to execute the synthetic data generation step. Depending on the input workload, `AIM` will privately release marginals that preserve certain linear queries more accurately. We find that a *two-way* marginal workload is sufficient for estimating or approximating the sufficient statistics for both linear and logistic regression. The proposed method is amenable to generalization beyond these classes of problems and can be potentially extended to others by i) identifying or approximating their sufficient statistics and ii) customizing the workload passed on as input in `AIM` accordingly. Since our proposed methods are based on post-processing DP workload query answers, the differential privacy analysis is straightforward (Definition 2.6).

### 3.1   Numerical encoding

We assume discrete (or discretized) input data, which is a common format for tabular data and is required by `AIM` and other marginal-based approaches. However, for machine learning, each record $\chi$ must be mapped to a numerical vector $z = (x, y)$ where $x \in \mathbb{R}^p$ is a feature vector and $y \in \mathbb{R}$ is a target. While the details of this encoding are often overlooked, they are important here for two reasons. First, they are needed to tightly bound $\|x\|$, which is used in sensitivity calculations of a number of DP ML methods, with tighter bounds leading to higher utility. Second, the encoding is a key part of recovering sufficient statistics from marginals.

Let $\psi_j(\chi_j) \in \mathbb{R}^{m_j}$ be the one-hot encoding of $\chi_j$, i.e., the vector with entries $\psi_{j,s}(\chi_j) = \mathbf{1}[\chi_j = s]$ for each $s \in \Omega_j$. We consider any numerical encoding of the form $\chi_j \mapsto A_j \psi_j(\chi_j)$ where $A_j \in \mathbb{R}^{p_j \times m_j}$ is a fixed linear transformation applied to the one-hot vector. This covers two special cases of interest. The first is the scalar encoding with $A_j = v_j^T$ for a vector $v_j \in \mathbb{R}^{m_j}$ that specifies the numerical value for each $s \in \Omega_j$. In this case the mapping simplifies to $\chi_j \mapsto v_j(\chi_j)$. The second special case of interest is when $A_j = I_j$ is the $m_j \times m_j$ identity matrix, so the mapping simplifies to $\chi_j \mapsto A_j(\chi_j)$ to give the one-hot encoding itself. Another common variation is a reduced one-hot encoding where $A_j = \tilde{I}_j \in \mathbb{R}^{(m_j-1) \times m_j}$ is equal to $I_j$ with one row dropped to avoid redundant information in the one-hot encoding. We include a simple example of the encoding strategy in Appendix B.

The full encoded record is $z = (z_{[j]})_{j=1}^d$ where $z_{[j]} = A_j \psi_j(\chi_j) \in \mathbb{R}^{p_j}$ is the encoding of the $j$th attribute and these column vectors are concatenated vertically. A single entry of $z$ is selected as the target variable $y$ leaving a feature vector $x$ of dimension $p := (\sum_{j=1}^d p_j) - 1$. Later, we will also use indexing expressions like $(\cdot)_{[j]}$ and $(\cdot)_{[j,k]}$ to refer to blocks of a vector or matrix corresponding to the encoding of the $j$th and $k$th attributes.

Let $z^{(i)} = (x^{(i)}, y^{(i)})$ denote the encoding of record $\chi^{(i)}$ and let $X \in \mathbb{R}^{n \times p}$ be the matrix with $i$th row equal to $(x^{(i)})^T$ and $y \in \mathbb{R}^n$ be the vector with $i$th entry equal to $y^{(i)}$. Many DP ML methods require bounds on the magnitude of the encoded data. Let $\|\mathcal{X}\| = \sup_{x \in \mathcal{X}} \|x\|$ and $\|\mathcal{Y}\| = \sup_{y \in \mathcal{Y}} |y|$ be bounds provided by the user where $\mathcal{X} \subset \mathbb{R}^p$ and $\mathcal{Y} \subset \mathbb{R}$ are guaranteed to contain all possible encoded feature vectors $x$ and target values $y$, respectively. For example, a typical bound is $\|\mathcal{X}\| = \|x^+\|$ where $x_k^+ \geq \sup |x_k|$ bounds the magnitude of a single feature. If $x_k$ is the scalar encoding of $\chi_j$ we can take $x_k^+ = \max_{s \in \Omega_j} |v_j(s)|$. The following proposition describes how to tightly bound a feature vector that combines scalar features and one-hot encoded features.

**Proposition 3.1.** *Suppose $x = (u, w)$ where $u \in \mathbb{R}^a$ satisfies $\|u\| \leq \|\mathcal{U}\|$ and $w \in \mathbb{R}^b$ contains the one-hot encodings (either reduced or not reduced) of $c$ attributes. Then $\|\mathcal{X}\| := \sqrt{\|\mathcal{U}\|^2 + c}$ is an upper bound on $\|x\|$.*

*Proof.* $\|x\|^2 = \|u\|^2 + \|w\|^2 \leq \|\mathcal{U}\|^2 + c$ where $\|w\|^2 \leq c$ because $w$ is the concatenation of $c$ vectors each with at most a single entry of 1 and all other entries equal to 0. $\qquad\square$

Suppose $u$ consists of scalar features and $\|\mathcal{U}\|$ is obtained by bounding each one separately as described above. This bound is tighter than the naive one of $\sqrt{\|\mathcal{U}\|^2 + b}$ that would be obtained by bounding each entry of the one-hot vectors separately.

### 3.2 Linear regression

The goal of linear regression is to minimize the sum of squared differences between the observed values $y$ and predicted values $X\theta$ in a linear model with $\theta \in \mathbb{R}^p$. The ordinary least squares (OLS) estimator is obtained by minimizing the squared error loss function $\|y - X\theta\|^2$. Mathematically, the OLS estimator is given by $\hat{\theta} = (X^T X)^{-1} X^T y$. In this context, the sufficient statistics are $T(X, y) = \{X^T X, X^T y\}$. In DD-SSP, we approximate $T(X, y)$ using linear queries. Specifically, we show that each entry of $X^T X$ and $X^T y$ can be obtained from pairwise marginals. The sufficient statistics we will consider all have the form of empirical second moments of the encoded attributes.

#### 3.2.1 Sufficient Statistics from Pairwise Marginals

Let $Z = [X, y] \in \mathbb{R}^{n \times (p+1)}$. The matrix $Z^T Z$ has blocks that contain our sufficient statistics of interest:

$$Z^T Z = \begin{bmatrix} X^T X & X^T y \\ y^T X & y^T y \end{bmatrix}. \tag{1}$$

However, we will see that we can also construct $Z^T Z$ directly from marginals.

**Proposition 3.2.** *Let $(Z^T Z)_{[j,k]}$ be the block of $Z^T Z$ with rows corresponding to the $j$th attribute encoding $z_{[j]} = A_j \psi_j(\chi_j)$ and columns corresponding to the $k$th attribute encoding $z_{[k]} = A_k \psi_k(\chi_k)$. Then*

$$(Z^T Z)_{[j,k]} = A_j \langle \mu_{j,k} \rangle A_k^T$$

*where $\langle \mu_{j,k} \rangle \in \mathbb{R}^{m_j \times m_k}$ is the $(j, k)$-marginal shaped as a matrix with $(s, t)$ entry $\mu_{j,k}[s, t] = \sum_{i=1}^n \mathbf{1}[\chi_j^{(i)} = s, \chi_k^{(i)} = t]$. Note that according to this definition $\langle \mu_{j,j} \rangle = diag(\mu_j)$.*

This shows that we can reconstruct the sufficient statistic matrix $Z^T Z$ directly from the set of all single-attribute and pairwise marginals. Note that single-attribute marginals $\mu_j$ can be constructed from any $\mu_{j,k}$ with $k \neq j$.

*Proof.* The sufficient statistic matrix can be written as $Z^T Z = \sum_{i=1}^n z^{(i)} (z^{(i)})^T$. Indexing by blocks gives

$$(Z^T Z)_{[j,k]} = \sum_{i=1}^n z_{[j]}^{(i)} (z_{[k]}^{(i)})^T$$

$$= \sum_{i=1}^{n} A_j \psi_j(\chi_j^{(i)}) \psi_k(\chi_k^{(i)})^T A_k^T$$

$$= A_j \Big( \sum_{i=1}^{n} \psi_j(\chi_j^{(i)}) \psi_k(\chi_k^{(i)})^T \Big) A_k^T$$

$$= A_j \langle \mu_{j,k} \rangle A_k^T,$$

In the last line, we used that $\psi_j(\chi_j^{(i)}) \psi_k(\chi_k^{(i)})^T$ is a matrix with $(s,t)$ entry equal to $\mathbf{1}[\chi_j^{(i)} = s, \chi_k^{(i)} = t]$, so summing over all $i$ gives the matrix $\langle \mu_{j,k} \rangle$. □

Algorithm 1 outlines how to retrieve approximate sufficient statistics $\widetilde{X^T X}$ and $\widetilde{X^T y}$ from marginals privately estimated by `AIM`. This implies we can solve DP linear regression by i) retrieving $\widetilde{X^T X}$ and $\widetilde{X^T y}$ as outlined in Algorithm 1, and ii) finding $\hat{\theta}_{\mathrm{DP}} = \widetilde{X^T X}^{-1} \widetilde{X^T y}$.

---

**Algorithm 1** `DD-SSP`

---

1: $M \leftarrow \mathtt{DPQuery}(\mathcal{D}, \epsilon, \delta)$ is the collection of privately computed pairwise marginal tables $\mu_{j,k}$ for all attribute pairs $(j,k)$, computed by DP query release algorithm of choice `DPQuery`.
2: $(\widetilde{Z^T Z})_{[j,k]} \leftarrow A_j \langle \mu_{j,k} \rangle A_k^T$ for all attribute pairs $(j,k)$ (see Proposition 3.2)
3: Extract $\widetilde{X^T X}$ and $\widetilde{X^T y}$ from $\widetilde{Z^T Z}$ using the block structure of Equation (1)

---

**Proposition 3.3.** *`DD-SSP` is $(\epsilon, \delta)$-DP.*

The proof follows directly from the $(\epsilon, \delta)$-DP properties of the marginal-releasing algorithm (in our case, `AIM`), and the fact that all subsequent steps are post-processing of a DP result (Definition 2.6).

### 3.3 Logistic regression

Logistic regression predicts the probability a binary label $y \in \{-1, +1\}$ takes value $+1$ as $p = 1/(1+\exp(-x \cdot \theta))$, where $\theta \in \mathbb{R}^p$ is a coefficient vector and $x \cdot \theta$ is the dot-product. The log-likelihood function is

$$\ell(\theta) = \sum_{i=1}^{n} \phi(x^{(i)} \cdot \theta y^{(i)})$$

where $\phi(s) := -\log(1 + e^{-s})$. Optimizing this log-likelihood is a convex optimization problem solvable numerically via standard optimizers. The log-likelihood does not have *finite* sufficient statistics. However, Huggins et al. (2017) offers a polynomial strategy to obtain *approximate* sufficient statistics for generalized linear models (GLMs), including logistic regression. Kulkarni et al. (2021) used a similar polynomial approximation for private Bayesian GLMs.

We propose a novel DP logistic regression method that combines two ideas: i) we use a Chebyshev approximation of the logistic regression log-likelihood based on Huggins et al. (2017), which allows us to write the objective in terms of approximate sufficient statistics, and then ii) use `AIM` privately released marginals to estimate the approximate sufficient statistics without accessing the sensitive data. This gives us the option to directly optimize an approximate log-likelihood based on privatized linear queries computed by `AIM`. The choice of the input workload for `AIM` depends on the characterization of the approximate log-likelihood. Based on our derivation below, we find that a suitable workload input for logistic regression is all pairwise marginals.

Huggins et al. (2017) propose to approximate $\ell(\theta)$ by using an degree-$M$ polynomial approximation of the function $\phi$:

$$\phi(s) \approx \phi_M(s) := \sum_{m=0}^{M} b_m^{(M)} s^m$$

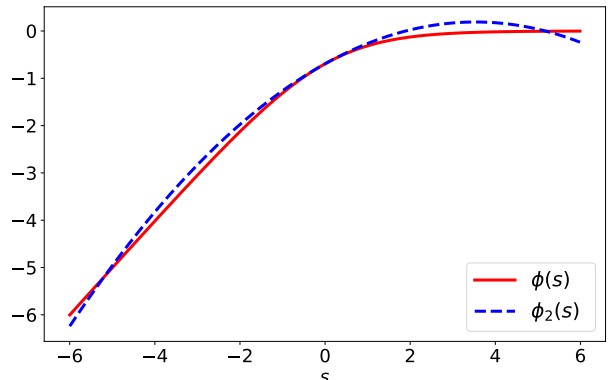

Figure 2: Degree 2 Chebyshev approximation of the logit function $\phi$, where $\phi(s) := -\log(1 + e^{-s})$. The inner products $\langle y^{(i)} x^{(i)}, \theta \rangle$ tend to be concentrated in the range $[-4, 4]$ across many datasets (Huggins et al., 2017). We conservatively choose range $[-6, 6]$ based on the inner DP products of the chosen datasets.

where $b_j^{(M)}$ are constants. There are different choices for the orthogonal polynomial basis, and as in Huggins et al. (2017), we focus on Chebyshev polynomials, which provide uniform quality guarantees over a finite interval $[-R, R]$ for positive $R$ (Figure 2). We can then write

$$\ell(\theta) \approx \sum_{i=1}^{n} \sum_{m=0}^{M} b_m^{(M)} (x^{(i)} \cdot \theta y^{(i)})^m$$

Based on the distribution of noisy inner products obtained through objective perturbation (Figure 4, Appendix C), we choose to work with a degree-2 Chebyshev approximation over the range $[-6, 6]$, leading to a precise approximation over a range that encompasses most of the observed inner product values.

**Proposition 3.4.** *The logistic regression log-likelihood is approximated by second order Chebyshev polynomial $\tilde{\ell}(\theta) \approx n b_0^{(2)} + b_1^{(2)} \theta \widetilde{X^T y} + b_2^{(2)} \cdot (\theta^T \widetilde{X^T X} \theta)$, where $b_0^{(2)}$, $b_1^{(2)}$, and $b_2^{(2)}$ are constants, and $\widetilde{X^T X}$ and $\widetilde{X^T y}$ can be retrieved from pairwise marginals as in Proposition 3.2. The maximizer of this approximate log-likelihood is available in closed form as $\hat{\theta}_{\text{DP-cheb}} = -\frac{b_1^{(2)}}{2\,b_2^{(2)}} (\widetilde{X^T X})^{-1} \widetilde{X^T y}$*

The proof is provided in Appendix C. This allows us to define a logistic regression objective where $\widetilde{X^T X}$ and $\widetilde{X^T y}$ are obtained via Algorithm 1, which is DP by post-processing (Proposition 3.3). Further, the optimal solution to this objective is a simple rescaling of the DD-SSP linear regression solution using these features and labels. It can be shown that the Chebyshev coefficients for $\phi(s)$ satisfy $b_1^{(2)} > 0$ and $b_2^{(2)} < 0$, so the rescaling factor $-\frac{b_1^{(2)}}{2\,b_2^{(2)}}$ is positive.

### 3.4 Connections to synthetic data

Based on the insight that sufficient statistics (or their approximations) can be expressed as linear queries, the proposed framework highlights a novel connection between differentially private synthetic data generation and SSP (Figure 1). Query answering algorithms are often used in synthetic data generation procedures. Many such procedures follow the select-measure-reconstruct approach to synthetic data, where linear queries are privately estimated from noisy measurements, and integrated into a model from which synthetic data can be sampled. This process ensures that the output synthetic data supports the selected linear queries. The synthetic data can then be used downstream to compute any statistic of interest, or in the example of linear regression, we could fit the model by training on the synthetic data. This workflow differs from DD-SSP only in that it consolidates the noisy measurements into a model and samples synthetic data from it.

By training select-measure-reconstruct synthetic generative models to preserve the appropriate workload of queries, synthetic data can therefore be "tuned" for specific machine learning tasks; for example, based on the findings in 3.2 and 3.3, we expect synthetic data that preserves pairwise marginals to perform well on linear and logistic regression, as it implicitly computes the relevant sufficient statistics (or approximate sufficient statistics). These findings provide a new perspective that enhances the understanding and utility of private synthetic data, especially as related to synthetic data for machine learning tasks.

### 3.5 Generality of the approach

The proposed approach is amenable to generalization on two fronts: i) the flexibility with respect to the choice of private linear query answering method, and ii) the applicability to a variety of models.

Query-answering methods are often used in synthetic data generation algorithms. Virtually any private query answering algorithm, such as Vietri et al. (2020); McKenna et al. (2021b); Aydore et al. (2021), could be adopted in the context of `DD-SSP`. To give a concrete example, we run supplementary experiments using `MST` (McKenna et al., 2021b) instead of `AIM` within our framework (Appendix H). `Private-PGM` can also be replaced with different methods for model estimation from the private measurements. In Appendix H, we show results for experiments where `AIM` is combined with the mixture inference step from `RAP` (Aydore et al., 2021).

The `DD-SSP` approach can be extended to any model where sufficient statistics can be linked to marginal queries either directly (as we demonstrate for linear regression in 3.2) or via a polynomial approximation (as in logistic regression, 3.3). More detailed extensions of the method are discussed in Section 5.

### 3.6 Baselines

We compare `DD-SSP` to synthetic data method `AIM-Synth`, where private measurements are used to estimate the underlying distribution, and surrogate data is sampled from it. We train `AIM-Synth` with an input workload of all pairwise marginals to match the workload utilized for `DD-SSP`. Since marginal-based synthetic data is designed to preserve the same linear queries that are sufficient to solve linear regression, or approximately sufficient for logistic regression, the expectation is that `DD-SSP` will closely match the performance of `AIM-Synth`. The difference between these approaches is whether linear queries for sufficient statistics are computed directly from marginals estimated by the mechanism, or computed from synthetic data (Figure 1).

We also compare both methods against established DP baselines. Since our methods are based on privately reconstructing sufficient statistics, for DP linear regression sufficient statistic perturbation (`SSP`) is the natural baseline choice. We choose `AdaSSP` (Wang, 2018) for its competitive performance. `AdaSSP` uses limited data-adaptivity to add a ridge penalty based on an estimated bound on the eigenvalues of $X^T X$, but then adds independent noise to each entry of the sufficient statistics, unlike fully data-adaptive query-answering mechanisms. The `AdaSSP` algorithm is detailed in Appendix D. For logistic regression, objective perturbation (`ObjPert`) is a widely adopted solution originally proposed by Chaudhuri et al. (2011), and further refined by Kifer et al. (2012) where it is extended to $(\epsilon, \delta)$-DP, with more general applicability and improved guarantees. Algorithm and details are provided in Appendix E. Both `AdaSSP` and `ObjPert` determine how much noise to add based on $\|\mathcal{X}\|$, which is the upper bound to the $L_2$-norm of any row of $X$ (Section 3.1). For example, in `AdaSSP`, $X^T X$ is noise-perturbed proportionally to $\|\mathcal{X}\|^2$. From Proposition 3.1, we can set $\|\mathcal{X}\|^2 = \|\mathcal{U}\|^2 + c$ where $\|\mathcal{U}\|$ is a bound on the numerically-encoded features and $c$ is the number of one-hot-encoded attributes to obtain a tight sensitivity bound for these baselines.

In addition to our proposed methods, we compare against `DP-SGD` (Abadi et al., 2016), a widely adopted algorithm for differentially private training. `DP-SGD` is highly sensitive to the choice of hyperparameters and requires extensive tuning to achieve optimal performance. To accurately account for the privacy loss incurred during hyperparameter tuning, advanced privacy accounting techniques must be applied (Ponomareva et al., 2023). To illustrate the variability in `DP-SGD`'s performance, our plots depict a shaded region representing the range between two versions of `DP-SGD`: an "optimistic" baseline that disregards the privacy cost of hyperparameter tuning, artificially inflating performance, and a more realistic version that incorporates this cost using advanced composition (Steinke, 2022). The effective performance of `DP-SGD` in practical scenarios,

including with more advanced privacy accounting (Papernot and Steinke, 2021), is expected to lie within this range.

### 3.7 Limitations

When choosing `AIM` as the mechanism for DP marginals, we are limited to working with discrete data, which is a requirement in `AIM` itself. Thus, our comparisons to other regression methods are scoped to discrete numerical data. Future work may consider `DD-SSP` with other mechanisms that support continuous data without discretization. For the logistic regression approximation, we use Chebyshev second-order polynomials; other approximation functions and/or degrees of precision could be evaluated. As shown in section G, `DD-SSP` demonstrates overall gains in regression accuracy compared to baseline methods, however this improvement comes at the cost of increased computational time tied to the complexity involved in privately releasing private marginals for large domains with high utility. This cost can be significantly reduced by replacing `AIM` with faster marginal-releasing methods, which we demonstrate in Appendix H. More detail on the computational runtime is discussed in Appendix G, including a few accessible strategies to mitigate the computational costs in real-world applications.

## 4 Experiments

In our experiments, we evaluate the effectiveness of `DD-SSP` on both linear and logistic regression tasks.[1] For linear regression, our main goal is to demonstrate the effectiveness of data-dependence for SSP. We therefore compare primarily against `AdaSSP`, a standard SSP baseline. For reference, we also compare to the public solution and to `DP-SGD` (Abadi et al., 2016), a widely used non-SSP method, as a representative of this class of methods; several other non-SSP approaches have been recently proposed for differentially private linear regression, including model selection via approximate Tukey depth (Amin et al., 2022), and iterative private gradient descent (Brown et al., 2024). For SSP, Tang et al. (2024) recently integrated gradient boosting and clipping with `AdaSSP` with the primary goal of mitigating the performance impacts due to data clipping when tight bounds on the range of the data are not known in advance; this is largely orthogonal to our goal of improving SSP via data-dependence in selecting and measuring queries. For logistic regression, we benchmark `DD-SSP` against `ObjPert` and `DP-SGD`. Both are widely adopted methods for logistic regression under differential privacy, with the caveat that `DP-SGD` has strong dependence on hyperparameter tuning — as we later further discuss. We also assess the performance of `AIM`-generated synthetic data (`AIM-Synth`), observing that it closely mirrors `DD-SSP` across tasks, reinforcing our hypothesis that data-dependent estimation of sufficient statistics underlies the strong performance of marginal-based synthetic data in machine learning settings.

We compare the Mean Squared Error (MSE) of DP query-based methods `DD-SSP` and `AIM-Synth` against the DP baseline `AdaSSP` and the public baseline for $\epsilon \in \{0.05, 0.1, 0.5, 1.0, 2.0\}$, with a fixed $\delta = 10^{-5}$. Figure 3 shows that `DD-SSP` and `AIM-Synth` have nearly identical performance and both improve significantly upon `AdaSSP` on all datasets except ACSIncome, where performance is similar. For logistic regression, `DD-SSP` closely matches `AIM-Synth` and surpasses `ObjPert` in low-$\epsilon$ regimes while being competitive at higher $\epsilon$ values. Assessing `DD-SSP` vs. `DP-SGD` is more challenging due to its reliance on hyperparameter fine-tuning. We represent this variability with a shaded region spanning the case where `DP-SGD` accounts for the privacy cost of hyperparameter tuning via advanced composition (Steinke, 2022), and the case where this cost is ignored. Our results show that `DD-SSP` is competitive with `DP-SGD` under advanced composition, with performance varying across datasets. Additional details on experimental setup, datasets, and implementation can be found in Appendix G.

Based on these results, we observe the following:

- `DD-SSP` is a competitive option for DP linear and logistic regression, surpassing data-independent SSP and `ObjPert` baselines in specific cases, and being competitive with `DP-SGD` overall.

---

[1]All experiment code is available at `https://github.com/ceciliaferrando/DD-SSP`.

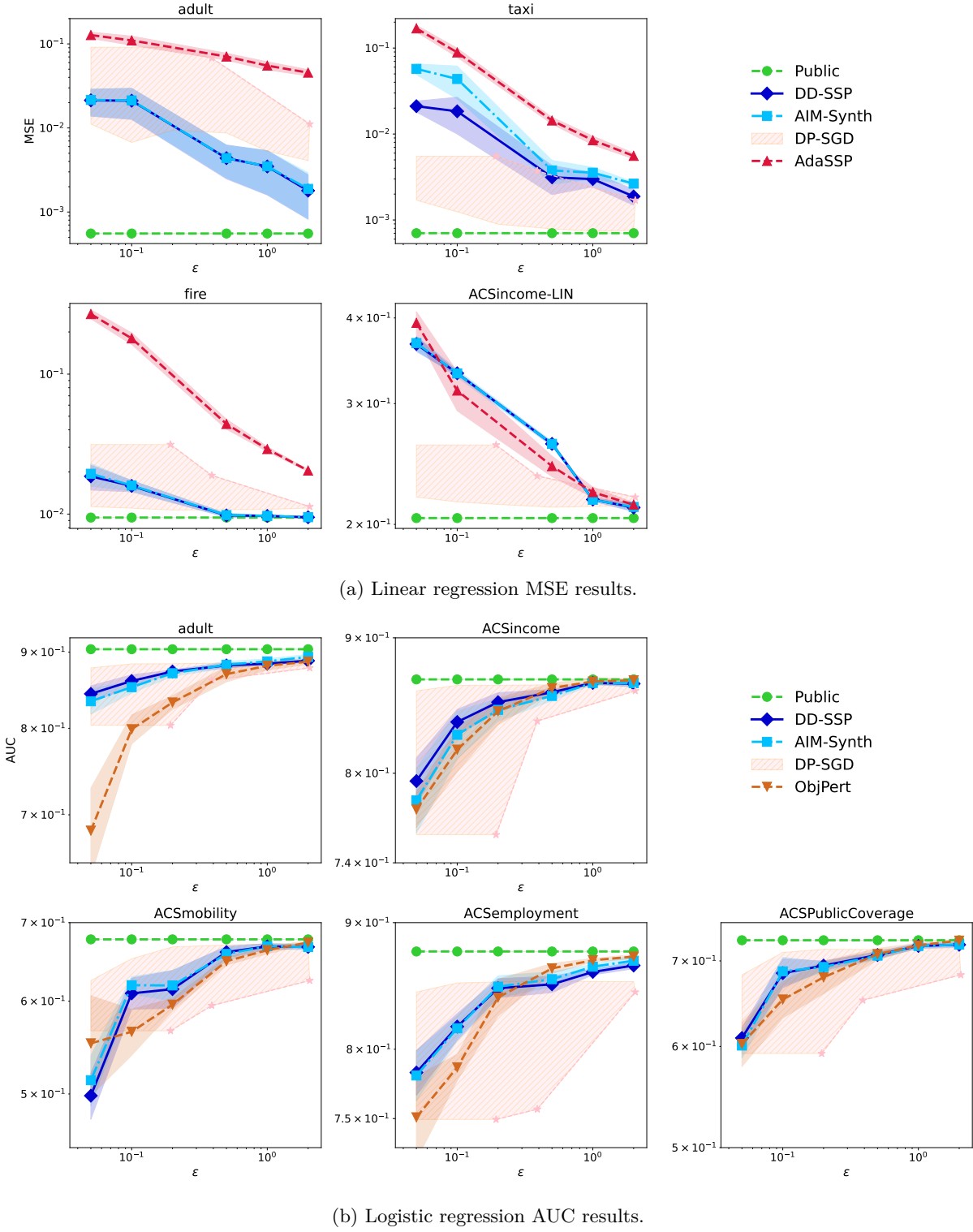

(a) Linear regression MSE results.

(b) Logistic regression AUC results.

Figure 3: Performance comparison across linear and logistic regression. Standard error bars are computed over 5 trials. For DP-SGD, the shaded region spans the range between an optimistic baseline ignoring hyperparameter tuning cost, and a realistic baseline using advanced composition to account for this cost (this line is highlighted by star markers). The advanced composition curve starts further to the right as the lowest $\epsilon$ achievable by DP-SGD in these cases is 0.005.

- The performance of `AIM-Synth` suggests that estimating problem-specific data-dependent sufficient statistics explains the suitability of `AIM` synthetic data for machine learning tasks. This implies `DD-SSP` is effective whenever pairwise marginals are available.
- The approximate `DD-SSP` method for logistic regression constitutes a novel DP algorithm as an alternative to privatized ERM procedures.

## 5 Future work

We imagine extensions to two sets of models. Beyond linear regression, all exponential family distributions, including graphical models like Naive Bayes, have finite sufficient statistics, and for such models we can devise similar `DD-SSP` solutions, or tailor synthetic data, by identifying a workload that supports the estimation of their sufficient statistics. Additionally, future work can focus on developing approximate loss functions with finite sufficient statistics for a broader class of other models, including generalized linear models (GLMs), where our results for logistic regression can be extended to obtain approximate sufficient statistics from $k$-way marginals by making a $k$-degree polynomial approximation to the GLM mapping function following the reasoning of Huggins et al. (2017). This will open the door to novel `DD-SSP` methods that directly minimize the approximate loss functions, and improve utility by adding privacy noise in a data-dependent way. Examples of such extensions are outlined in Appendix F. Additionally, methods targeting encoded workload $A_j \langle \mu_{j,k} \rangle A_k^T$ instead of $\mu_{j,k}$ can be explored and combined with advanced data-independent mechanisms like the matrix mechanism, potentially leading to a new class of encoding-aware `SSP` methods.

## 6 Conclusions

We introduce methods for data-dependent sufficient statistic perturbation (`DD-SSP`). Our methods use privately released marginal tables to solve linear and logistic regression via sufficient statistics. We find that `DD-SSP` performs better than data-independent SSP on linear regression and objective perturbation for logistic regression, and is competitive with `DP-SGD`, known to achieve excellent results under fine-tuned hyperparameter setting.

Notably, the approximate `DD-SSP` logistic regression algorithm is the first DP logistic regression method that allows analysts to solve logistic regression via a SSP algorithm, directly minimizing the approximate loss function. Additionally, we find that the performance of `DD-SSP` is almost indistinguishable from that of `AIM` synthetic data: this suggests that with the appropriate workload, training these machine learning models on query-based DP synthetic data corresponds to data-dependent SSP.

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

# Appendix

## A  AIM

For `AIM`, we follow the original algorithm in McKenna et al. (2022). `AIM` uses an intelligent initialization step to estimate one-way marginals. This results in a model where all one-way marginals are preserved well, and higher-order marginals can be estimated under an independence assumption. Additionally, `AIM` uses a carefully chosen subset of the marginal queries and leverages the observation that lower-dimensional marginals exhibit a better signal-to-noise ratio than marginals with many attributes and low counts, and at the same time they can be used to estimate higher-dimensional marginal queries in the workload. Finally, the quality score function for selecting marginals to measure ensures that the selection is "budget-adaptive", i.e. it measures larger dimensional marginals only when the available privacy budget is large enough.

---

**Algorithm 2** AIM (McKenna et al., 2022)

---

1: **Input:** Dataset $\mathcal{D}$, workload $W$, privacy parameter $\rho$
2: **Output:** Synthetic Dataset $\tilde{\mathcal{D}}$
3: **Hyper-Parameters:** MAX-SIZE=80MB, $T = 16d$, $\alpha = 0.9$
4: $\sigma_0 = \sqrt{T/(2\,\alpha\,\rho)}$
5: $\rho_{used} = 0$
6: $t = 0$
7: Initialize $\hat{p}_t$ (using Algorithm 3)
8: $w_r = \sum_{s \in W} c_s \mid r \cap s \mid$
9: $\sigma_{t+1} \leftarrow \sigma_0 \quad \epsilon_{t+1} \leftarrow \sqrt{8(1-\alpha)\rho/T}$
10: **while** $\rho_{used} < \rho$ **do**
11: $\quad t = t + 1$
12: $\quad \rho_{used} \leftarrow \rho_{used} + \frac{1}{8}\epsilon_t^2 + \frac{1}{2\sigma_t^2}$
13: $\quad C_t = r_t \in W_+ \mid \text{JunctionTree-SIZE}(r_1, \ldots, r_t))$
$\qquad \leq \frac{\rho_{used}}{\rho} \cdot \text{MAX-SIZE}$
14: $\quad$ **select** $r_t \in C_t$ using the exponential mechanism with:

$$q_r(\mathcal{D}) = w_r \Big( \|M_r(\mathcal{D}) - M_r(\hat{p}_{t-1})\|_1 - \sqrt{2/\pi} \cdot \sigma_t \cdot n_r \Big)$$

15: $\quad$ **measure** marginal on $r_t$:
$$\tilde{y}_t = M_{r_t}(\mathcal{D}) + \mathcal{N}(0, \sigma_t^2 I)$$

16: $\quad$ **estimate** data distribution using `Private-PGM`:

$$\hat{p}_t = \arg\min_{p \in S} \sum_{i=1}^{t} \frac{1}{\sigma_i} \|M_{r_i}(p) - \tilde{y}_i\|_2^2$$

17: $\quad$ anneal $\epsilon_{t+1}$ and $\sigma_{t+1}$ using Algorithm 4
18: **end while**
19: **generate** synthetic data $\tilde{\mathcal{D}}$ from $\hat{p}_t$ using `Private-PGM`
20: **return** $\tilde{\mathcal{D}}$

---

---

**Algorithm 3** Initialize $p_t$ (Subroutine of Algorithm 2) (McKenna et al., 2022)

1: **for** $r \in \{r \in W_+ \mid |r| = 1\}$ **do**
2:      $t \leftarrow t + 1$
3:      $\sigma_t \leftarrow \sigma_0$
4:      $r_t \leftarrow r$
5:      $\tilde{y}_t \leftarrow M_r(\mathcal{D}) + \mathcal{N}(0, \sigma_t^2 \mathbb{I})$
6:      $\rho_{\text{used}} \leftarrow \rho_{\text{used}} + \frac{1}{2\sigma_t^2}$
7: **end for**
8: $\hat{p}_t \leftarrow \text{argmin}_{p \in S} \sum_{i=1}^{t} \frac{1}{\sigma_i} \|M_{r_i}(p) - \tilde{y}_i\|_2^2$

---

**Algorithm 4** Budget Annealing (Subroutine of Algorithm 2) (McKenna et al., 2022)

1: **if** $\|M_{r_t}(\hat{p}_t) - M_{r_t}(\hat{p}_{t-1})\|_1 \leq \sqrt{\frac{2}{\pi}} \cdot \sigma_t \cdot n_{r_t}$ **then**
2:      $\epsilon_{t+1} \leftarrow 2 \cdot \epsilon_t$
3:      $\sigma_{t+1} \leftarrow \sigma_t / 2$
4: **else**
5:      $\epsilon_{t+1} \leftarrow \epsilon_t$
6:      $\sigma_{t+1} \leftarrow \sigma_t$
7: **end if**
8: **if** $(\rho - \rho_{\text{used}}) \leq 2 \left( \frac{1}{2\sigma_{t+1}^2} + \frac{1}{8}\epsilon_{t+1}^2 \right)$ **then**
9:      $\epsilon_{t+1} \leftarrow \sqrt{8 \cdot (1 - \alpha) \cdot (\rho - \rho_{\text{used}})}$
10:      $\sigma_{t+1} \leftarrow \sqrt{\frac{1}{2 \cdot \alpha \cdot (\rho - \rho_{\text{used}})}}$
11: **end if**

---

# B Encoding example

The following example demonstrates the application of the encoding strategy in 3.1. The mapping is:

$$\chi_j \mapsto A_j \psi_j(\chi_j)$$

For a concrete example, suppose there are 5 levels (i.e., $m_j = 5$) and the feature value is $\chi_j = 3$, then the one-hot encoding vector is

$$\psi_j(\chi_j) = \begin{bmatrix} 0 \\ 0 \\ 1 \\ 0 \\ 0 \end{bmatrix}$$

For the numerical encoding we use $A_j = v_j^T$. For example, suppose $v_j^T = [1, 2, 4, 8, 16]$, then in our example we have

$$\chi_j \mapsto v_j^T \psi_j(\chi_j) = [1, 2, 4, 8, 16] \begin{bmatrix} 0 \\ 0 \\ 1 \\ 0 \\ 0 \end{bmatrix} = 4$$

It is easy to see that the numerical value is always equal to $v_j[\chi_j]$, i.e., value in vector $v_j$ at index $\chi_j$. In other words, the vector $v_j$ enumerates the numerical values for each level. For the one-hot encoding case, we use $A_j = I_j$, the identity matrix. In our example, this gives

$$\chi_j \mapsto I_{m_j} \psi_j(\chi_j) = \psi_j(\chi_j) = \begin{bmatrix} 0 \\ 0 \\ 1 \\ 0 \\ 0 \end{bmatrix}$$

It is clear that this gives the one-hot encoding. The reduced one-hot encoding is similar.

## C  Logistic Regression log-likelihood approximation

*Proof.* The log-likelihood for logistic regression can be expressed as

$$\ell(\theta) = \sum_{i=1}^{n} \phi(x^{(i)} \cdot \theta y^{(i)})$$

where $\phi(s) := -\log(1 + e^{-s})$. Based on Huggins et al. (2017), we can approximate the logistic regression log-likelihood with a Chebyshev polynomial approximation of degree $M$:

$$\phi(s) \approx \phi_M(s) := \sum_{m=0}^{M} b_m^{(M)} s^m$$

where $b_j^{(M)}$ are constants. Then,

$$\ell(\theta) \approx \sum_{i=1}^{n} \sum_{m=0}^{M} b_m^{(M)} (x^{(i)} \cdot \theta y^{(i)})^m$$

If we choose $M = 2$,

$$\ell(\theta) \approx \sum_{i=1}^{n} b_0^{(2)} + b_1^{(2)} \cdot (x^{(i)} \cdot \theta y^{(i)}) + b_2^{(2)} \cdot (x^{(i)} \cdot \theta y^{(i)})^2$$

The quadratic term is

$$\sum_{i=1}^{n} (x^{(i)} \cdot \theta y^{(i)})^2 = \sum_{i=1}^{n} y^{(i)^2} \sum_{j,k=1}^{d} x_j^{(i)} x_k^{(i)} \theta_j \theta_k = \sum_{j,k=1}^{d} \theta_j \theta_k \sum_{i=1}^{n} y^{(i)^2} x_j^{(i)} x_k^{(i)}$$

Therefore we can rewrite the approximate log-likelihood as

$$\ell(\theta) \approx n b_0^{(2)} + b_1^{(2)} \sum_{i=1}^{n} x^{(i)} \cdot \theta y^{(i)} + b_2^{(2)} \sum_{j,k=1}^{d} \theta_j \theta_k \sum_{i=1}^{n} y^{(i)^2} x_j^{(i)} x_k^{(i)}$$

$$\approx n b_0^{(2)} + b_1^{(2)} y^T X \theta + b_2^{(2)} \sum_{j,k=1}^{d} \theta_j \theta_k \sum_{i=1}^{n} x_j^{(i)} x_k^{(i)}$$

$$\approx n b_0^{(2)} + b_1^{(2)} y^T X \theta + b_2^{(2)} \theta^T X^T X \theta$$

where the simplification in the second line follows from the fact that we work with $y^{(i)} \in \{1, -1\}$ and $y^{(i)^2} = 1$ for any $i$.

After moving constants, an equivalent objective for maximization w.r.t. $\theta$ is:

$$\tilde{\ell}(\theta) \ \propto \ \frac{b_2^{(2)}}{b_1^{(2)}} \theta^T X^T X \theta \ + \ \theta^T X^T y \ + \ \text{const.}$$

For comparison, the linear-regression log-likelihood under a Gaussian noise model is:

$$\ell_{\text{linear}}(\theta) = -\frac{1}{2} \theta^T X^T X \theta \ + \ \theta^T X^T y \ + \ \text{const.}$$

It can be shown that the Chebyshev coefficients for $\phi(s)$ satisfy $b_1^{(2)} > 0$ and $b_2^{(2)} < 0$, so both functions are concave and quadratic, with terms having the same signs, and the only difference being the relative weighting of the quadratic and linear terms. A simple derivation (set the gradient of $\tilde{\ell}$ to zero and solve for $\theta$ assuming $\widetilde{X^T X}$ is invertible) shows that $\tilde{\ell}(\theta)$ has the following minimizer, which is a positive rescaling of the OLS solution:

$$\hat{\theta}_{\text{cheb}} = -\frac{b_1^{(2)}}{2\,b_2^{(2)}}\,(X^T X)^{-1} X^T y.$$

The sufficient statistics $X^T X$ and $X^T y$ can be derived following the same proposed strategies as in linear regression (Section 3.2), obtaining marginal query-based estimates $\widetilde{X^T X}$ and $\widetilde{X^T y}$. We can then express the closed-form solution of the approximate log-likelihood with approximate sufficient statistics as:

$$\hat{\theta}_{\text{DP-cheb}} = -\frac{b_1^{(2)}}{2\,b_2^{(2)}}\,(\widetilde{X^T X})^{-1}\widetilde{X^T y}.$$

$\square$

Figure 4 shows the distribution of the inner DP products for the datasets used in the experiments, supporting the choice of $[-6, 6]$ for the range of the Chebyshev approximation.

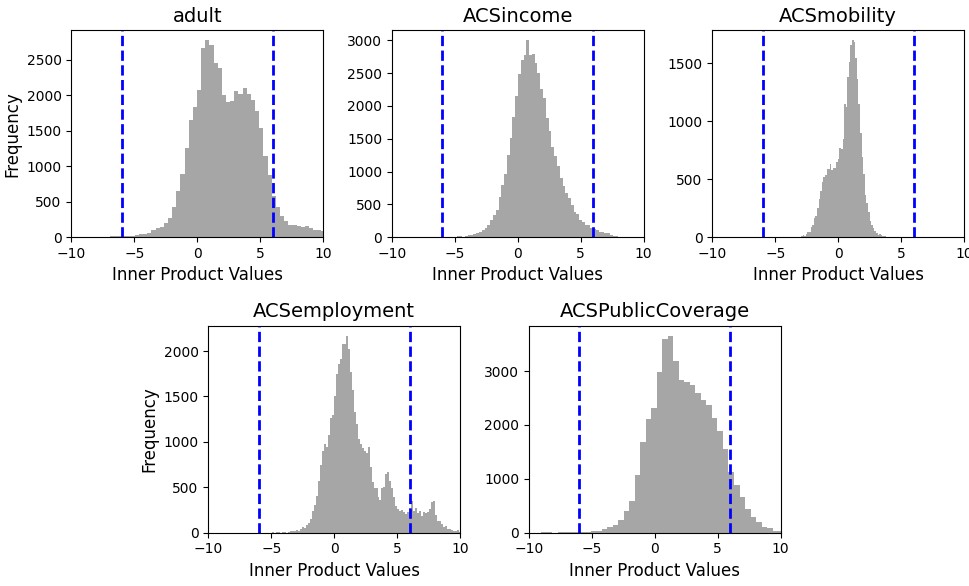

Figure 4: Distribution of the inner DP products $\langle y^{(i)} x^{(i)}, \hat{\theta}_{DP} \rangle$ for the datasets of interest. $\hat{\theta}_{DP}$ is computed using the objective perturbation method with $\epsilon = 1$ and $\delta = 10^{-5}$. The dashed vertical lines represent the [-6, 6] bounds chosen for the Chebyshev approximation.

# D  Linear regression baseline

Algorithm 5 outlines the `AdaSSP` method for linear regression (Wang, 2018).

---

**Algorithm 5** AdaSSP (Wang, 2018)

---

1: **Input:** Data $X$, $y$. Privacy budget: $\epsilon$, $\delta$. Bounds: $\|\mathcal{X}\|$, $\|\mathcal{Y}\|$, $\rho \in (0,1)$ (0.05 in the paper)

2: **1.** Calculate the minimum eigenvalue $\lambda_{\min}\left(X^T X\right)$.

3: **2.** Privately release $\tilde{\lambda}_{\min} = \max\left\{\lambda_{\min} + \frac{\sqrt{\log(6/\delta)}}{\epsilon/3}\|\mathcal{X}\|^2 Z - \frac{\log(6/\delta)}{\epsilon/3}\|\mathcal{X}\|^2, 0\right\}$, where $Z \sim \mathcal{N}(0,1)$.

4: **3.** Set $\lambda = \max\left\{0, \frac{\sqrt{d\log(6/\delta)\log(2d^2/\rho)}\|\mathcal{X}\|^2}{\epsilon/3} - \tilde{\lambda}_{\min}\right\}$.

5: **4.** Privately release $\widehat{X^T X} = X^T X + \frac{\sqrt{\log(6/\delta)}\|\mathcal{X}\|^2}{\epsilon/3} Z$ for $Z \in \mathbb{R}^{d \times d}$ is a symmetric matrix and every element from the upper triangular matrix is sampled from $\mathcal{N}(0,1)$.

6: **5.** Privately release $\widehat{X^T y} = X^T y + \frac{\sqrt{\log(6/\delta)}\|\mathcal{X}\|\|\mathcal{Y}\|}{\epsilon/3} Z$ for $Z \sim \mathcal{N}\left(0, I_d\right)$.

7: **Output:** $\tilde{\theta} = \left(\widehat{X^T X} + \lambda I\right)^{-1} \widehat{X^T y}$.

---

To reason about the sensitivity of $X^T X$, consider two neighboring datasets $X \in \mathbb{R}^{n \times d}$ and $X' \in \mathbb{R}^{(n+1) \times d}$ differing by one data entry $v \in \mathcal{X}$, where $v$ is a $d \times 1$ vector. Then,

$$\Delta_{X^T X} = \sup_{X \sim X'} \|f(X') - f(X)\|_F$$

Since $X$ and $X'$ only differ by one row $(v)$, then $f(X') - f(X) = vv^T$ (Sheffet, 2017).

So the sensitivity is maximum over $v$ of $\|\text{vec}(vv^T)\| = \|vv^T\|_F$. We have

$$\begin{aligned}
\Delta_{X^T X}^2 &= \sup_{v \in \mathcal{X}} \|vv^T\|_F^2 \\
&= \sup_{v \in \mathcal{X}} \sum_{i=1}^{d} \sum_{j=1}^{d} (v_i v_j)^2 \\
&= \sup_{v \in \mathcal{X}} \left(\sum_{i=1}^{d} v_i^2\right)\left(\sum_{j=1}^{d} v_j^2\right) \\
&= \sup_{v \in \mathcal{X}} \|v\|^4 \\
&= \|\mathcal{X}\|^4
\end{aligned}$$

where $\|\mathcal{X}\|$ is the greatest possible norm of a vector in the domain $\mathcal{X}$. Therefore,

$$\Delta_{X^T X} = \|\mathcal{X}\|^2.$$

The sensitivity of $X^T y$ can be similarly derived. Given neighboring datasets $X \in \mathbb{R}^{n \times d}, y \in \mathbb{R}^n$, and $X' \in \mathbb{R}^{(n+1) \times d}, y' \in \mathbb{R}^{n+1}$, where $v \in \mathcal{X} \subset \mathbb{R}^d$ is the new row, and $w \in \mathcal{Y} \subset \mathbb{R}$ is the new value in $y'$. Then,

$$\|f\left(X', y'\right) - f(X, y)\| = \|X'^T y' - X^T y\| = \|wv\|$$

Since $\|\mathcal{X}\| = \sup_{x \in \mathcal{X}} \|x\|$ and $\|\mathcal{Y}\| = \sup_{y \in \mathcal{Y}} |y|$, we have

$$\begin{aligned}
\Delta_{X^T y} &= \sup_{(X,y) \sim (X',y')} \|f\left(X', y'\right) - f(X, y)\| \\
&= \sup_{w \in \mathcal{Y}, v \in \mathcal{X}} |w| \cdot \|v\| = \|\mathcal{Y}\| \cdot \|\mathcal{X}\|
\end{aligned}$$

# E    Logistic regression baseline

Our DP logistic regression baseline is based on the generalized objective perturbation algorithm in Kifer et al. (2012) (Algorithm 6). In this section, to match the notation in Kifer et al. (2012), $\ell(\theta; z) = \log(1 + \exp(-x \cdot \theta y))$ is the loss for a single datum and $\hat{\mathcal{L}}(\theta; \mathcal{D}) = \frac{1}{n} \sum_{i=1}^{n} \ell(\theta; z^{(i)})$ is the average loss over the dataset.

---

**Algorithm 6** Generalized Objective Perturbation Mechanism (`ObjPert`) (Kifer et al., 2012)

---

**Require:** dataset $\mathcal{D} = \{z^{(1)}, \ldots, z^{(n)}\}$, where $z^{(i)} = (x^{(i)}, y^{(i)})$, privacy parameters $\epsilon$ and $\delta$ ($\delta = 0$ for $\epsilon$-differential privacy), bound $\|\mathcal{X}\|$ on the $L_2$ norm of any $x$ entry, convex regularizer $r$, a convex domain $\mathbb{F} \subseteq \mathbb{R}^d$, convex loss function $\hat{\mathcal{L}}(\theta; \mathcal{D}) = \frac{1}{n} \sum_{i=1}^{n} \ell(\theta; z^{(i)})$, with continuous Hessian, $\|\nabla \ell(\theta; z)\| \leq \|\mathcal{X}\|$ (for all $z \in \mathcal{D}$ and $\theta \in \mathbb{F}$), and the eigenvalues of $\nabla^2 \ell(\theta; z)$ bounded by $\frac{\|\mathcal{X}\|^2}{4}$ (for all $z$ and for all $\theta \in \mathbb{F}$).

1: Set $\Delta \geq \frac{\|\mathcal{X}\|^2}{2\epsilon}$.

2: Sample $b \in \mathbb{R}^d$ from $\nu_2(b; \epsilon, \delta, \|\mathcal{X}\|) = \mathcal{N}\left(0, \frac{\|\mathcal{X}\|^2 \left(8 \log \frac{2}{\delta} + 4\epsilon\right)}{\epsilon^2} I_d\right)$.

3: $\hat{\theta}_{\mathrm{DP}} \equiv \arg\min_{\theta \in \mathbb{F}} \hat{\mathcal{L}}(\theta; \mathcal{D}) + \frac{1}{n} r(\theta) + \frac{\Delta}{2n} \|\theta\|^2 + \frac{b^T \theta}{n}$.

---

The algorithm requires the following bounds for the gradient and Hessian of $\ell$:

$$\|\nabla \ell(\theta; z)\| \leq \|\mathcal{X}\|$$

$$\lambda_{\max}(\nabla^2 \ell(\theta; z)) \leq \frac{\|\mathcal{X}\|^2}{4}$$

To reason about the sensitivity bounds, let $\phi(s) = \log(1 + e^s)$. Then we can write $\ell(\theta; z) = \phi(-x \cdot \theta y)$. Following Gower and Bach (2019), it is straightforward to derive that

$$\phi'(s) = \frac{e^s}{1 + e^s} \leq 1$$

$$\phi''(s) = \frac{e^s}{(1 + e^s)^2} \leq \frac{1}{4}$$

and clear that both quantities are non-negative. Then the gradient of $\ell$ is:

$$\nabla \ell(\theta; z) = \nabla_\theta \phi(-x \cdot \theta y) = \phi'(-x \cdot \theta y) \cdot -yx$$

The norm is bounded as

$$\|\nabla \ell(\theta; z)\| = |\phi'(-x \cdot \theta y)| \cdot |y| \cdot \|x\| \leq \|\mathcal{X}\|$$

where the inequality holds since $|\phi'(s)| \leq 1$ for all $s$ and $|y| = 1$.

By differentiating the gradient again and using the fact that $y^2 = 1$, we can derive the Hessian as:

$$\nabla^2 \ell(\theta; z) = \phi''(-x \cdot \theta y) x x^T$$

The maximum eigenvalue is

$$\begin{aligned}
\lambda_{\max}(\nabla^2 \ell(\theta; z)) &= \phi''(-x \cdot \theta y) \lambda_{\max}(x x^T) \\
&= \phi''(-x \cdot \theta y) \|x\|^2 \\
&\leq \frac{\|\mathcal{X}\|^2}{4}
\end{aligned}$$

In the second line, we used the fact that $\lambda_{\max}(x x^T) = \|x\|^2$. To see this, note that $x x^T$ is rank one and $(x x^T) x = \|x\|^2 x$, therefore $x$ is an eigenvector with eigenvalue $\|x\|^2$ and this is the largest eigenvalue. In the last line we used that $\|\phi''(s)\| \leq \frac{1}{4}$ for all $s$ and that $\|x\| \leq \|\mathcal{X}\|$.

# F    Examples of Extensions to GLMs and Higher-Order Approximations

## F.1    Degree-$k$ Polynomials from $k$-way Marginals

In this Section, we show that in general a loss function approximated by a degree-$k$ polynomial can be computed from the set of all $k$-way marginals.

Consider a loss function $\ell(\theta) = \sum_{i=1}^{n} \phi(x^{(i)}, \theta)$ where $\phi(x, \theta)$ is a degree-three polynomial in $x$. By breaking the polynomial into monomials, the loss function can be broken into a sum of terms, a generic one of which is $\sum_{i=1}^{n} C \cdot x_a^{(i)} x_b^{(i)} x_c^{(i)}$ for some constant $C$ and indices $a, b, c$ into the encoded feature vector. Recall that encoded features arise from linear transformations of one-hot encoded attributes via the mappings $\chi_j \mapsto A_j \psi_j(\chi_j)$. Suppose features $x_a, x_b, x_c$ come from attributes $\chi_j, \chi_k, \chi_\ell$, respectively. Then there are constant vectors $u \in \mathbb{R}^{m_j}$, $v \in \mathbb{R}^{m_k}$, $w \in \mathbb{R}^{m_\ell}$ (each a row of $A_j, A_k, A_\ell$, respectively) such that

$$x_a = u^\top \psi_j(\chi_j)$$
$$x_b = v^\top \psi_k(\chi_k)$$
$$x_c = w^\top \psi_\ell(\chi_\ell)$$

We can now rewrite the term from the loss function (ignoring the constant) as

$$
\begin{aligned}
\sum_{i=1}^{n} x_a^{(i)} x_b^{(i)} x_c^{(i)} &= \sum_{i=1}^{n} \left( u^\top \psi_j(\chi_j^{(i)}) \right) \left( v^\top \psi_k(\chi_k^{(i)}) \right) \left( w^\top \psi_\ell(\chi_\ell^{(i)}) \right) \\
&= \sum_{i=1}^{n} \left( \sum_{r=1}^{m_j} u_r \mathbb{I}[\chi_j^{(i)} = r] \right) \left( \sum_{s=1}^{m_k} v_s \mathbb{I}[\chi_k^{(i)} = s] \right) \left( \sum_{t=1}^{m_\ell} w_t \mathbb{I}[\chi_\ell^{(i)} = t] \right) \\
&= \sum_{i=1}^{n} \sum_{r=1}^{m_j} \sum_{s=1}^{m_k} \sum_{t=1}^{m_\ell} u_r v_s w_t \mathbb{I}[\chi_j^{(i)} = r] \mathbb{I}[\chi_k^{(i)} = s] \mathbb{I}[\chi_\ell^{(i)} = t] \\
&= \sum_{r=1}^{m_j} \sum_{s=1}^{m_k} \sum_{t=1}^{m_\ell} u_r v_s w_t \sum_{i=1}^{n} \mathbb{I}[\chi_j^{(i)} = r, \chi_k^{(i)} = s, \chi_\ell^{(i)} = t] \\
&= \sum_{r=1}^{m_j} \sum_{s=1}^{m_k} \sum_{t=1}^{m_\ell} u_r v_s w_t \cdot \langle \mu_{j,k,\ell} \rangle_{rst} \\
&= \langle u \otimes v \otimes w, \langle \mu_{j,k,\ell} \rangle \rangle
\end{aligned}
$$

In the second-to-last line, we introduced the notation $\langle \mu_{j,k,\ell} \rangle \in \mathbb{R}^{m_j \times m_k \times m_\ell}$ for the array containing the three-dimensional data marginal, with entries

$$\langle \mu_{j,k,\ell} \rangle_{rst} = \sum_{i=1}^{n} \mathbb{I}[\chi_j^{(i)} = r, \chi_k^{(i)} = s, \chi_\ell^{(i)} = t].$$

In the last line, we rewrote the sum as an inner product between the constant tensor $u \otimes v \otimes w$, derived from the feature encoding, and the marginal $\langle \mu_{j,k,\ell} \rangle$, to emphasize the general form of the operation. More generally, a loss function term that depends on at most $k$ features, like a degree-$k$ monomial, can be rewritten as a tensor inner product between a constant tensor and a $k$-way marginal. Thus, if the overall loss function is a degree-$k$ polynomial, it can be computed from the set of all $k$-way marginals.

## F.2    Poisson Regression (Degree-2 Chebyshev Approximation, Log-Link)

Consider a Poisson GLM with canonical log-link function:

$$y^{(i)} \sim \text{Poisson}(\mu^{(i)}), \quad \eta^{(i)} = \log(\mu^{(i)}) = x^{(i)} \cdot \theta.$$

The standard Poisson regression log-likelihood is:

$$\ell(\theta) = \sum_{i=1}^{n} \left[ y^{(i)}(x^{(i)} \cdot \theta) - e^{x^{(i)} \cdot \theta} - \log(y^{(i)}!) \right].$$

Ignoring the constant term $\log(y^{(i)}!)$ (which does not depend on $\theta$), we have:

$$\ell(\theta) \propto \sum_{i=1}^{n} \left[ y^{(i)}(x^{(i)} \cdot \theta) - e^{x^{(i)} \cdot \theta} \right].$$

If we approximate the exponential function $e^s$ using a second-degree Chebyshev polynomial approximation around a chosen interval (e.g. $[-6, 6]$) as

$$e^s \approx c_0^{(2)} + c_1^{(2)} s + c_2^{(2)} s^2,$$

where $c_0^{(2)}, c_1^{(2)}, c_2^{(2)}$ are Chebyshev polynomial coefficients determined numerically, then, substituting this approximation into the Poisson log-likelihood gives:

$$\tilde{\ell}(\theta) \approx \sum_{i=1}^{n} \left[ y^{(i)}(x^{(i)} \cdot \theta) - \left( c_0^{(2)} + c_1^{(2)}(x^{(i)} \cdot \theta) + c_2^{(2)}(x^{(i)} \cdot \theta)^2 \right) \right].$$

Expanding explicitly and simplifying, we obtain:

$$\tilde{\ell}(\theta) = (X^T y)^T \theta - c_0^{(2)} n - c_1^{(2)} \mathbf{1}^T X \theta - c_2^{(2)} \theta^T (X^T X) \theta,$$

where $X^T X$, $X^T y$, and $\mathbf{1}^T X$ can be computed from all pairwise marginals (see Section F.1).

### F.3 Logistic Regression (Degree-4 Chebyshev Approximation)

The logistic regression log-likelihood (with labels $y^{(i)} \in \{-1, 1\}$) is:

$$\ell(\theta) = \sum_{i=1}^{n} \log \left( \frac{1}{1 + e^{-x^{(i)} \cdot \theta y^{(i)}}} \right) = -\sum_{i=1}^{n} \log \left( 1 + e^{-x^{(i)} \cdot \theta y^{(i)}} \right).$$

We approximate the logistic function $\phi(s) = -\log(1 + e^{-s})$ using a degree-4 Chebyshev polynomial:

$$\phi(s) \approx \sum_{m=0}^{4} b_m^{(4)} s^m,$$

where the constants $b_m^{(4)}$ are determined numerically by fitting Chebyshev polynomials over a chosen interval (e.g. $[-6, 6]$). Substituting this approximation into the log-likelihood, we obtain:

$$\tilde{\ell}(\theta) \approx \sum_{i=1}^{n} \sum_{m=0}^{4} b_m \left( x^{(i)} \cdot \theta y^{(i)} \right)^m.$$

Expanding explicitly, this yields:

$$\tilde{\ell}(\theta) \approx b_0 n + b_1 \theta^T (X^T y) + b_2 \theta^T (X^T X) \theta + b_3 \sum_{i=1}^{n} y^{(i)} (\theta^T x^{(i)})^3 + b_4 \sum_{i=1}^{n} (\theta^T x^{(i)})^4.$$

The required sufficient statistics are $X^T y$, $X^T X$, $\sum_{i=1}^{n} y^{(i)} (x^{(i)})^{\otimes 3}$, and $\sum_{i=1}^{n} (x^{(i)})^{\otimes 4}$. These can be computed from 4-way marginals (see Section F.1).

Note that measuring higher-order marginals under differential privacy poses both computational and statistical challenges. From a computational standpoint, the domain size of a marginal grows exponentially with its order, making both private measurement and post-processing increasingly expensive. Statistically, higher-order marginals tend to suffer from lower signal-to-noise ratios, as the added noise (due to higher sensitivity and larger output space) can overwhelm the utility of the measurement. As noted in the `AIM` paper (McKenna et al., 2022), lower-order marginals often provide a better trade-off, and can still be leveraged to estimate higher-order interactions via structured inference methods, such as graphical models.

## G  Experiment details

### G.1  Datasets and Preprocessing

We use the following datasets:[2]

- **Adult** (Becker and Kohavi, 1996): The target variable is 'num-education' (number of education years) for linear regression and 'income>50K' for logistic regression.

- **Fire** (Ridgeway et al., 2021): The target variable is 'Priority' (of the call).

- **Taxi** (Grégoire et al., 2021): The target variable is 'totalamount' (total fare amount).

- **ACS Datasets** (Ding et al., 2021): Data is queried for California (2018). Includes binary classification tasks for 'PINCP' (income above \$50k), 'MIG' (mobility), 'ESR' (employment), and 'PUBCOV' (public coverage). **ACSincome** is also used for linear regression with the target variable 'PINCP' (income) discretized into 20 bins.

More detail on the datasets is included in Table 1. Data is shuffled and split into 1,000 test points and up to 50,000 training points. Non-numerical features are one-hot encoded, dropping the first level to avoid multi-collinearity, and numerical features are rescaled to $[-1, 1]$ for noise allocation (see Section 3.1 for more detail on data encoding).

### G.2  Methodology

**`AIM` training:** `AIM` is trained with a model size of 200MB, a maximum of 1,000 iterations, and a workload of all pairwise marginals. For `AdaSSP`, sensitivity is calibrated as described in Sections 3.1 and 3.6.

**`DP-SGD` fine tuning and training:** `DP-SGD`'s hyperparameters are fine-tuned by running a gridsearch for the best parameter. The search space spans the following values:

- Batch size: $[n, 1024, 256]$

- Gradient clipping norm: $[0.01, 0.1, 0.2]$

- Number or epochs: $[1, 10, 20]$

- Learning rate: $[0.001, 0.01, 0.1, 1.0]$

Advanced composition is used to account for hyperparameter tuning costs as per Theorem 22 in Steinke (2022). The optimistic baseline ignores this cost entirely.

---

[2]The ACS data is sourced from `https://github.com/socialfoundations/folktables`. All other datasets are sourced from `https://github.com/ryan112358/hd-datasets`.

Table 1: Dataset information

| Dataset | Size | # Attributes | Attributes | Target |
|---------|------|--------------|------------|--------|
| Adult | 48,842 | 15 | ['age', 'workclass', 'fnlwgt', 'education', 'marital-status', 'occupation', 'relationship', 'race', 'sex', 'capital-gain', 'capital-loss', 'hours-per-week', 'native-country', 'income>50K', 'education-num'] | 'income>50K' (logistic), 'education-num' (linear) |
| ACSIncome | 195,665 | 9 | ['AGEP', 'COW', 'SCHL', 'MAR', 'RELP', 'WKHP', 'SEX', 'RAC1P', 'PINCP'] | 'PINCP' |
| Fire | 305,119 | 15 | ['ALS Unit', 'Battalion', 'Call Final Disposition', 'Call Type', 'Call Type Group', 'City', 'Final Priority', 'Fire Prevention District', 'Neighborhooods - Analysis Boundaries', 'Original Priority', 'Station Area', 'Supervisor District', 'Unit Type', 'Zipcode of Incident', 'Priority'] | 'Priority' |
| Taxi | 1,048,575 | 11 | ['VendorID', 'passengercount', 'tripdistance', 'RatecodeID', 'PULocationID', 'DOLocationID', 'paymenttype', 'fareamount', 'tipamount', 'tollsamount', 'totalamount'] | 'totalamount' |
| ACSmobility | 29,358 | 20 | ['AGEP', 'SCHL', 'MAR', 'SEX', 'DIS', 'CIT', 'MIL', 'ANC', 'WKHP', 'NATIVITY', 'RELP', 'DEAR', 'DEYE', 'DREM', 'RAC1P', 'GCL', 'COW', 'ESR', 'JWMNP', 'PINCP'] | 'MIG' |
| ACSEmployment | 378,817 | 17 | ['AGEP', 'SCHL', 'MAR', 'RELP', 'DIS', 'ESP', 'CIT', 'MIG', 'MIL', 'ANC', 'NATIVITY', 'DEAR', 'DEYE', 'DREM', 'SEX', 'RAC1P', 'ESR'] | 'ESR' |
| ACSPublicCoverage | 138,550 | 19 | ['AGEP', 'SCHL', 'MAR', 'SEX', 'DIS', 'ESP', 'CIT', 'MIG', 'MIL', 'ANC', 'NATIVITY', 'DEAR', 'DEYE', 'DREM', 'PINCP', 'ESR', 'FER', 'RAC1P', 'PUBCOV'] | 'PUBCOV' |

### G.3 Computational runtime

All experiments were conducted on an internal cluster equipped with Xeon Gold 6240 CPUs @ 2.60GHz, 192GB RAM, and 240GB local SSD storage. The runtime of our method is influenced by several factors, including dataset size, domain size, and the privacy parameter $\epsilon$. Since our method relies on private marginals released by `AIM`, its runtime is inherently tied to `AIM`'s computational demands, which are significantly higher than those of the DP baselines.

For clarity, we focus the detailed runtime analysis on the Adult dataset as it is representative of general trends observed across all datasets. On this dataset, `AIM` runtime increases significantly with $\epsilon$. For linear regression, `AIM` requires approximately 8 minutes at $\epsilon = 0.05$, scaling up to 18 hours at $\epsilon = 2.0$. In comparison, the DP baseline `AdaSSP` completes the same experiment in approximately 5 seconds regardless of $\epsilon$. Similar trends are observed for logistic regression, with `AIM` runtime increasing from 12 minutes to 21 hours as $\epsilon$ grows, while the corresponding DP baseline `ObjPert` completes these experiments in approximately 2 seconds.

For `DP-SGD`, the runtime of the hyperparameter search phase varies across datasets, tasks and $\epsilon$ between around 1 hour and 17 hours; once the best hyperparameters have been determined, the runtime of `DP-SGD`, taking the case of linear regression on the Adult dataset as example, varies between 2 minutes for $\epsilon = 0.05$ and 10 minutes for $\epsilon = 2.0$.

### G.4 Runtime mitigation strategies

While AIM's runtime is substantial, it is a direct consequence of the complexity involved in accurately computing private marginals for large domains and high utility. A few factors mitigate the runtime of the proposed methods: i) `AIM` is much faster for low $\epsilon$ values; ii) `AIM` model size can be reduced to 100MB for a significant runtime cut, without significantly sacrificing the accuracy of the method. In the context of this paper, we choose to prioritize accuracy, which is consistent with the motivation of synthetic data, where computation is spent up front to release a data set that can be used downstreams in many ways; iii) other faster query answering mechanisms can be used in place of `AIM`, such as `MST` (see 3.5), which runs in approximately 18 minutes for a full experiment across all $\epsilon$ values on Adult. Various ways to improve the accuracy vs running-time trade-offs for use in specific settings can be explored, which is beyond the scope of this paper. In terms of experimentation (see Section 4), for computational viability we study the variability of our results across 5 trials.

# H  Additional experiments

`DD-SSP` can accommodate different DP marginal query releasing methods. For our main results we use `AIM` (see Appendix A), which uses `Private-PGM` (McKenna et al., 2019) for the "generate" step. In this section, we briefly demonstrate the use of the `DD-SSP` framework with alternative methods. In particular, we consider two alternative settings: i) replacing `AIM` with `MST` (McKenna et al., 2021b); ii) replacing the `Private-PGM` step in `AIM` with `MixtureInference`[3], a close approximation of the relaxed projection methods in (Aydore et al., 2021) and (Liu et al., 2021). Results are shown in Figure 5 and Figure 6 respectively.

From the ablation studies in (McKenna et al., 2022), we expect `MST` to perform well at low $\epsilon$ values (high privacy) and `AIM` to outperform it at higher $\epsilon$ values. `MST` performs a domain compression operation, and we hypothesize this benefits some cases (e.g. low $\epsilon$), but hurts in others. Based on the same ablation studies, we expect `Private-PGM` to yield lower workload errors with respect to `MixtureInference`, resulting in better overall metric scores for the tasks of interest. Consistently with the findings in (McKenna et al., 2022), we find that `AIM` using `Private-PGM` has the most reliable performance overall.

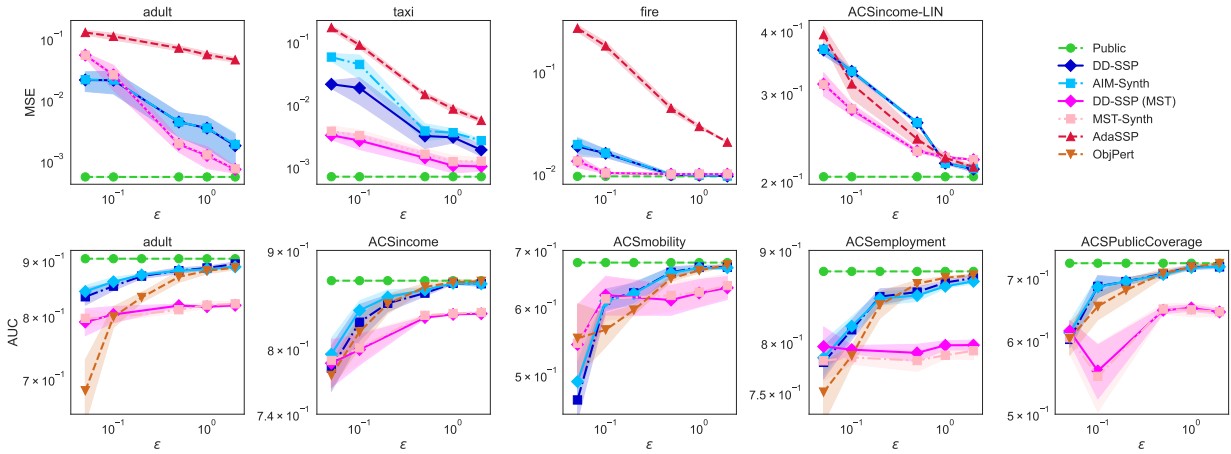

Figure 5: `MST` (McKenna et al., 2021b) vs. `AIM` (McKenna et al., 2022) as a query answering algorithm. Top: linear regression. Bottom: logistic regression. Standard error bars are computed over 5 trials.

---

[3]https://github.com/ryan112358/private-pgm/blob/master/src/mbi/mixture_inference.py

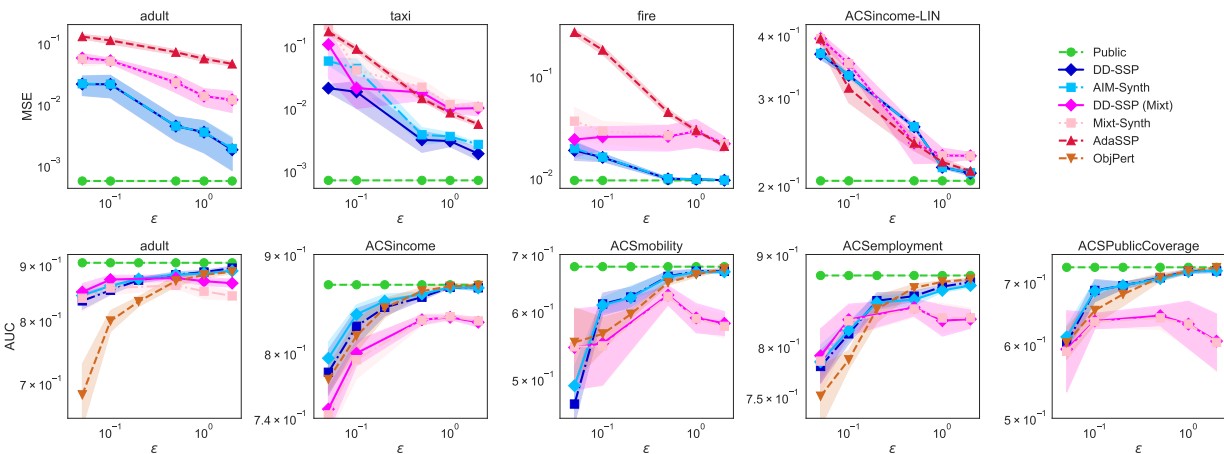

Figure 6: `AIM` using `MixtureInference` (Aydore et al., 2021; Liu et al., 2021) vs. `Private-PGM` (McKenna et al., 2019) for model estimation and synthetic data generation. Top: linear regression. Bottom: logistic regression. Standard error bars computer over 5 trials.

