# OpenReview forum: "Private Regression via Data-Dependent Sufficient Statistic Perturbation"
_TMLR — Accepted by TMLR_

### Review · Reviewer_jNnR · 2025-04-28

**Summary Of Contributions:**

This paper introduces Data-Dependent Sufficient Statistic Perturbation (DD-SSP), a new framework for differentially private regression analysis. It departs from traditional data-independent sufficient statistic perturbation (SSP) methods by leveraging private, data-dependent linear query answering to estimate sufficient statistics more accurately. Algorithms for both linear regression and logistic regression is derived. Experiments show that DD-SSP outperforms existing SSP methods and is competitive with fine-tuned DP-SGD baselines, particularly when accounting for the privacy cost of hyperparameter tuning. Also, the proposed framework is extentive to other models with either exact or approximate sufficient statistics.

**Audience:**

Yes

**Broader Impact Concerns:**

No concerns. This is a theoretical work that may have societal impacts in the future.

**Claims And Evidence:**

Yes

**Requested Changes:**

- Could the authors provide a utility analysis or at least evidence supporting why the proposed query answering scheme is superior to previous approaches? A privacy-utility trade-off discussion would be valuable.
- Could the authors comment on the challenges of approximating high-order polynomials using linear queries?
- Please expand the literature review to include relevant works. Also, is it possible to evaluate it in the experiments?
- I am curious about the difference between DD-SSP and AIM-Synth in the experimental results. Most of the time, their performances are almost exactly aligned, but there are noticeable gaps on certain datasets. What causes these gaps, given that the authors stated that "marginal-based synthetic data is designed to preserve the same linear queries that are sufficient to solve linear regression"? Also, the original AIM-Synth involves a marginal selection step. When DD-SSP and AIM-Synth exhibit similar performance, does this imply that the selection step does not waste any privacy budget?


Minor points:

- In the caption of Figure 1, the subfigures are referred to using "top" and "bottom," which may be a typo.
- Please ensure the correct usage of \cite, \citep, and \citet. Note that the formatting of citations may vary depending on the LaTeX template used.
- There is a typo in the first line of Section 5.

**Strengths And Weaknesses:**

Strengths:

- The framework is easy to extend in terms of query answering scheme.
- The motivation and logical flow are clear and well explained. The idea of using queries to approximate sufficient statistics seems quite intuitive.
- The experimental results show consistent improvements over existing methods.

Weaknesses:

- No utility analysis is provided.
- The current analysis only covers simple models of linear regression and logistic regression, and there is no evident generalization towards more complex model. Based on the polynomial approximation, it appears that high-order polynomials may sometimes be necessary for other machine learning problems. I suggest that the authors add a comment regarding the challenges of approximating high-order polynomials using linear queries.
- Some relevant literature, such as [1,2,3], which proposes tailored solutions for differentially private linear regression, is missing.
- In the experiments, the method ObjPert uses $\delta=0$, which is in fact not a fair comparison and should be stated clearly.


[1] Amin K, Joseph M, Ribero M, et al. Easy Differentially Private Linear Regression[C], The Eleventh International Conference on Learning Representations.

[2] Tang, S., Aydore, S., Kearns, M., Rho, S., Roth, A., Wang, Y., ... & Wu, Z. S. (2024, April). Improved differentially private regression via gradient boosting. In 2024 IEEE Conference on Secure and Trustworthy Machine Learning (SaTML) (pp. 33-56). IEEE.

[3] Brown, G., Dvijotham, K., Evans, G., Liu, D., Smith, A., & Thakurta, A. (2024). Private gradient descent for linear regression: Tighter error bounds and instance-specific uncertainty estimation. arXiv preprint arXiv:2402.13531.

---

> ### Author Response · Authors · 2025-06-02
>
> We thank the reviewer for the questions and comments, which we will address in order:
>
> “Weaknesses”, Q1: **The proposed framework builds conceptually on the utility guarantees by supporting query-answering methods**. While the AIM paper itself does not have separate utility proofs, it builds upon MWEM [1], which provides utility results. Similarly to how AIM presents practical improvements based on established methods with known utility results, this paper’s focus is on the introduction of a flexible framework for SSP with good practical performance and ease of use. For evidence supporting the advantage of data-dependent methods, we refer to the literature on data-dependent query-answering, such as [2] and [3].
>
> [1] M. Hardt, K. Ligett, and F. McSherry. 2012. A Simple and Practical Algorithm for Differentially Private Data Release. In Advances in Neural Information Processing Systems 25: 26th Annual Conference on Neural Information Processing Systems. 2012
>
> [2] C. Li, M. Hay, G. Miklau, & Y. Wang. A data-and workload-aware algorithm for range queries under differential privacy. arXiv preprint arXiv:1410.0265. 2014
>
> [3] R. McKenna, D. Sheldon, & G. Miklau. Graphical-model based estimation and inference for differential privacy. In International Conference on Machine Learning (pp. 4435-4444). PMLR. 2019
>
> “Weaknesses”, Q2: **We have included a dedicated Appendix exemplifying how the framework can be extended to other GLMs as well as higher-degree approximations**, specifying the type of workload required in each case. Here we also comment on the challenge of computing higher order marginals, which mainly boils down to computational costs. See Appendix F in updated draft.
>
> “Weaknesses”, Q3: **We thank the reviewer for these helpful pointers and have incorporated the suggested references into the discussion in Section 4**. While these are all valid differentially private linear regression methods, we focus on AdaSSP specifically because it is a purely SSP-based approach and has performed well in prior evaluations. We would like to emphasize that the main goal of this paper is to demonstrate that data-dependent approaches can improve SSP. For context, we have also included a comparison to DP-SGD, another empirically successful method. It is beyond the scope of this paper (and not needed to support the claims) to evaluate every private OLS method.
>
> “Weaknesses”, Q4: We use the $(\epsilon, delta)$ version of ObjPert proposed by Kifer et al., 2012 (Algorithm 6, Appendix E). **In our experiments, $\texttt{ObjPert}$ is evaluated with the same fixed $\delta$ as all other methods ($\delta = 10^{-5}$)**. This ensures a fair comparison under the same $(\epsilon, \delta)$-DP setting.
>
> “Requested changes”, Q1: Please refer to our response “Weaknesses”, Q1.
>
> “Requested changes”, Q2: Please refer to our response “Weaknesses”, Q2 and the new Appendix F.
>
> “Requested changes”, Q3: We have added the suggested references (see the response to “Weaknesses”, Q3.
>
> “Requested changes”, Q4: We thank the reviewer for the opportunity to expand on this. The marginal selection step is shared across AIM-Synth and DD-SSP (because DD-SSP post-processes private marginals from AIM) – therefore this cannot be the source of the differences. For linear regression, the only difference should be the **additional small amount of error when generating synthetic data** by sampling from the model distribution. This introduces an error of order $1/\sqrt{n}$. For logistic regression, the **AIM-Synth method uses the true logistic regression loss function with synthetic data, while we use the approximate loss function**, which introduces an additional difference.
>
> “Minor points”: **We thank the reviewer for catching these typos and inconsistencies – we’ve addressed all of them in the updated draft**.

---

### Review · Reviewer_YXx9 · 2025-05-15

**Summary Of Contributions:**

The submission introduces a novel data-dependent sufficient statistic perturbation (DD-SSP) method for differentially private linear and logistic regression. Unlike traditional data-independent SSP, DD-SSP leverages privately released marginals to estimate sufficient statistics, achieving improved utility for linear regression compared to state-of-the-art baselines like AdaSSP. For logistic regression, the authors propose a Chebyshev polynomial approximation to derive approximate sufficient statistics, enabling a competitive SSP-based approach. Experimental results demonstrate that DD-SSP outperforms AdaSSP for linear regression and is competitive with objective perturbation and DP-SGD for logistic regression, particularly in low-$\epsilon$ regimes.

**Audience:**

Yes

**Claims And Evidence:**

Yes

**Requested Changes:**

Mandatory changes :

- Clarify the relation with data generation and how you problem articulates around it.
- Explain the relations (if any) between the solution of the degree-2 Chebyshev approximation for logistic regression and the solution of the ordinary least-squares problem.

Minor problems :

- Definition 2.4 should be a Definition-Proposition
- Definition 2.5 should be a Proposition

**Strengths And Weaknesses:**

Strengths :

- Despite the approach essentially consisting in combining existing blocks, I like its generality.
- The authors compare DD-SSP against strong baselines (AdaSSP, ObjPert, DP-SGD) across multiple datasets, providing robust evidence of its performance.
- The framework’s flexibility to accommodate different query-answering algorithms and its potential extension to other models (e.g., exponential family distributions) suggest a strong and easy applicability in practice.

Weaknesses :

- The writing could be more concise and clear. The introduction of synthetic data generation early in the paper feels tangential to the main focus on regression, potentially confusing readers about the paper’s primary contribution. If I understand correctly, the general idea that the authors are trying to convey is that from a private data generator or private information of the data distribution, it should be possible to solve many downstream tasks by post-processing. However, as it is currently stated in the article, one might understand that the
- The notations are often overly intricate, which may hinder accessibility. Simplifying mathematical expressions and definitions could improve readability.
- The plots are difficult to interpret due to overlapping curves, particularly for DP-SGD, and the absence of ObjPert in some logistic regression plots is not explained.
- The choice of a degree-2 Chebyshev approximation for logistic regression leads to an optimization problem resembling a quadratic form, but the authors do not discuss its implications, such as the relationship to linear regression solutions.

---

> ### Author Response · Authors · 2025-06-02
> **Response to Reviewer YXx9**
>
> We thank the reviewer for recognizing the generality and practical applicability of the proposed method, and for the constructive feedback. We will respond point by point.
>
> “Weaknesses”, Q1: We thank the reviewer for this feedback, and we agree that the section on synthetic data could be streamlined and better positioned with respect to the focus of the paper – leveraging query-answering methods for data-dependent SSP. **Following this recommendation, we have revised this section (see 2.3 in the updated draft)**.
>
> “Weaknesses”, Q2: We will take one more pass to see if there is any extraneous notation, but on the whole we feel we have used the minimum amount of notation to be precise. One area that is actually rather technical and gets glossed over in many treatments, but is quite important, is the feature encoding. This is central to our method for extracting sufficient statistics from marginals, where careful bookkeeping is essential for correctness and generalization.
>
> “Weaknesses”, Q3: **We have made the plots bigger for improved readability**. ObjPert does indeed appear in each logistic regression panel.
>
> “Weaknesses”, Q4: Thank you for this comment. This provides an interesting opportunity to further elaborate on the interpretation of the proposed approximate logistic regression objective.
>
> The proposed degree-2 Chebyshev approximation leads to the following log-likelihood
> $$
> \tilde{\ell}(\theta)
> \approx n b_0^{(2)} + b_1^{(2)} \theta^T X^T y + b_2^{(2)} \theta^T X^T X \theta
> $$
> with $b_2^{(2)} < 0$, so this is a concave quadratic function (see Figure 2). After moving constants, we have the following equivalent expression for maximization with respect to $\theta$:
> $$
> \tilde \ell(\theta) \propto \tfrac{b_2^{(2)}}{b_1^{(2)}} \theta^T X^T X \theta + \theta^T X^T y + \text{const.}
> $$
> For comparison, we write the linear regression log-likelihood function under a Gaussian noise model:
> $$
> \ell_{\text{linear}}(\theta) = -\frac{1}{2} \theta^T X^T X \theta + \theta^T X^T y + \text{const}.
> $$
> So, **the difference is only the relative weighting of the terms in the objective**.
>
> **This brings up a point we missed previously but which is obvious in hindsight**. Since $\tilde \ell(\theta)$ is also concave and quadratic it has the following closed form maximizer, which has parameters that are proportional to the OLS solution:
> $$
> \theta^* = -\tfrac{b_1^{(2)}}{2b_2^{(2)}} (X^T X)^{-1} X^T y
> $$
> We previously solved this using a numerical optimizer. We didn't have time to do this for the rebuttal, but **in our final submission we would confirm experimentally that this gives identical results**.
>
> “Requested Changes”: **We have made the requested mandatory changes (please refer to “Weaknesses”, Q1 and Q4)**. We have also added a dedicated Proposition to formally state the $(\epsilon, \delta)$-DP guarantees of the Gaussian mechanism defined in Definition 2.4. We believe that overall these changes improve the clarity and precision of the discussion, and we remain open to additional comments.

---

### Review · Reviewer_1Mct · 2025-05-19

**Summary Of Contributions:**

This paper proposes a **data-dependent Sufficient Statistic Perturbation** (DD-SSP) for differentially private linear regression by post-processing privately released marginals, achieving improved accuracy over traditional data-independent SSP. A key extension is logistic regression by designing an approximate objective expressible via sufficient statistics, leading to a novel and competitive SSP method. Furthermore, this paper also establishes a connection between SSP and synthetic data generation, showing that for models with sufficient statistics, training on synthetic data corresponds to data-dependent SSP, with utility determined by the accuracy of answering associated linear queries.

**Audience:**

Yes

**Claims And Evidence:**

Yes

**Requested Changes:**

1. Some citation formats in the paper are improperly placed. For instance, on Page 2, the phrase “AIM McKenna et al. (2022)” should be formatted as “AIM (McKenna et al., 2022).” The authors are encouraged to review the entire manuscript to ensure citation style consistency throughout.

2. The full name of AIM is not provided in the paper. It would improve clarity to define this acronym upon its first use.

3. In Figure 3, it should be (left) and (right) instead of (top) and (bottom). The M in Figure 1 is also confusion. It would be better to provide its definition since DP marginals can mean a lot of things.

4. The details of AIM should be given in the main text for enhancing clarity.

5 On page 8, the reference seems wrong. There is no Figure 2.1 in this paper.

**Strengths And Weaknesses:**

1. The authors demonstrate that the data-dependent SSP (DD-SSP) outperforms the data-independent SSP (DI-SSP). This result is **intuitive** since having knowledge of the data distribution generally enables better design of differential privacy mechanisms. However, the authors should clarify under what specific assumptions or types of data distributions their proposed method achieves significant improvements.

2. The current paper seems focus on discrete data, which entails encoding for numericalization. What if the dataset contains numerical and ordinal data? The current method seems restricted to categorical data.


3. The authors employ polynomial approximation for logistic regression. Naturally, it can be also extended to generalized linear models. The authors should add some discussions on this for increasing the contributions of this paper.

4. The current paper fails to compare DI-SSP and DD-SSP in details. At least, I do not see how the proposed method differ in DI-SSP. The authors should prepare an additional section for highlighting the difference.

---

> ### Author Response · Authors · 2025-06-02
> **Response to Reviewer 1Mct**
>
> We thank the reviewer for the insightful comments. We will address questions and requests point by point.
>
> “Strengths and Weaknesses”, Q1: AIM and other data-dependent query answering methods are applicable to a wide range of data and **don’t make particular assumptions about data-generating distributions**. They do build underlying models of the data internally, which is what allows them to save privacy budget, so they may behave better for datasets that are good matches to these models, but there are no particular assumptions and, with enough privacy budget, these models can perform well on any dataset. To provide specific examples of such models:
> - Private-PGM [1] uses maximum entropy reconstructions, therefore it works better when there are near-conditional independence properties.
> - RAP [2] is a mixture of products method that works well under a different type of assumptions.
> - GEM [3] uses generative models parameterized by neural networks.
>
> “Strengths and Weaknesses”, Q2: **Our method supports datasets containing both categorical and ordinal** (discrete and numerical) **attributes**. As described in Appendix G.1, the ordinal and numerical features (which are not subject to one-hot encoding) are rescaled to the range [−1,1]. This facilitates accurate sensitivity calibration. In summary, while categorical attributes are one-hot encoded, ordinal features are encoded linearly. We agree that this detail is important and will move the relevant explanation from the appendix to the main text for clarity. We also want to emphasize that **data discretization is not only common in differential privacy, but often practical and beneficial [4]**. The main reason is that histograms have low sensitivity, as each entry contributes at most one count to one bin, making the application of mechanisms like the Laplace and Gaussian mechanism easy to calibrate and efficient. [4] presents in depth discussion on the best strategies and benefits of discretization.
>
> “Strengths and Weaknesses”, Q3: We thank the reviewer for the opportunity to expand on the generality of the method. **We have added an Appendix section (Appendix F) presenting examples of generalization, including a degree-2 Chebyshev approximation for Poisson regression**. We hope this addition provides readers with a more concrete understanding of how the method can be extended to other models.
>
> “Strengths and Weaknesses”, Q4: We agree with the reviewer that the distinction between DI-SSP and DD-SSP, which in the previous version we discussed under Introduction and Methods, deserves further highlighting in the paper. To improve clarity, **we have included a dedicated paragraph** in the updated draft (see Section 2.2). In summary, DI-SSP adds independent noise to sufficient statistics without regard to the data distribution, whereas DD-SSP estimates those same statistics from differentially private marginals released by data-dependent query answering algorithms such as AIM. The new section concisely explains why traditional SSP is data-independent, further highlighting its differences from DD-SSP. We remain open to additional feedback by the reviewer.
>
> "Requested Changes": **Thank you for pointing these out – we have made the requested changes**.
>
> [1] R. McKenna, D. Sheldon, and G. Miklau. “Graphical-model based estimation and inference for differential privacy”. In International Conference on Machine Learning, pages 4435–4444. PMLR, 2019
>
> [2] S. Aydore, W. Brown, M. Kearns, K. Kenthapadi, L. Melis, A. Roth, and A. A. Siva. “Differentially private query release through adaptive projection”. In International Conference on Machine Learning, pages 457–467. PMLR, 2021
>
> [3] T. Liu, G. Vietri, and Z. S. Wu. "Iterative methods for private synthetic data: Unifying framework and new methods". In Advances in Neural Information Processing Systems 34: 690-702. 2021
>
> [4] G. Ganev, M. Annamalai, S. Mahiou, and E. De Cristofaro. "The Importance of Being Discrete: Measuring the Impact of Discretization in End-to-End Differentially Private Synthetic Data". arXiv preprint arXiv:2504.06923. 2025

---

### Author Response · Authors · 2025-06-02

We'd like to thank all reviewers for their constructive feedback and questions. We’ve carefully addressed each point in our responses and have revised the paper draft accordingly. All updated sections are highlighted in blue for clarity.

We look forward to continued discussion and are happy to address any further questions or suggestions.

---

### Decision · Action_Editor_11Ly · 2025-06-27

**Recommendation:** Accept as is

**Additional Comments:**

For the camera-ready, authors could consider addressing the following point that caught my eye while reading the paper: In Sec. 2.2. you outline the DI-SSP approach and state that noise is scaled to global sensitivity. It seems that you describe perturbing both entries of the suff. stat. tuple with noise scaled to the global sensitivity of the _particular_ _entry_. I guess this would technically require splitting the epsilon in half in order to allow make the total privacy budget $(\epsilon, \delta)$? This is a minor detail but I think it should be clarified in order to DI-SSP approach better.

However, my decision is "accept as is" and hence the change is not mandatory, but something I think would improve the clarity.

**Audience:**

Yes

**Audience Explanation:**

This paper studies differentially private linear regression model, which is an important model for many practical applications. Furthermore, the data dependent perturbation is an interesting subtopic for the differential privacy community. Hence I believe there is definitely an audience for this paper amongst the TMLR audience.

**Claims And Evidence:**

Yes

**Claims Explanation:**

In this paper, authors study differential private learning of linear and logistic regression through data dependent sufficient statistic perturbation (SSP). The proposed approach is based on learning a synthetic data generator for the sensitive data under DP, and compute the sufficient statistics from the synthetic data. Data independent SSP uniformly adds noise to the different parts of the sufficient statistics, hence possibly leading to sub-optimal signal-to-noise in some parts of the sufficient statistic vectors.

Authors demonstrate empirically, that the proposed approach outperforms the previous SSP approach significantly in the linear regression task. For the logistic regression task, authors demonstrate that the proposed approach outperform objective perturbation, and performs on par with the DP-SGD.

All the reviewers, as well as I, were satisfied with the evidence provided for the claims.